# SymProFold: Structural prediction of symmetrical biological assemblies

Christoph Buhlheller [1,2,8], Theo Sagmeister [1,8], Christoph Grininger [1], Nina Gubensäk [1], Uwe B. Sleytr[3], Isabel Usón[4,5] & Tea Pavkov-Keller [1,6,7] ✉

Symmetry in nature often emerges from self-assembly processes and serves a wide range of functions. Cell surface layers (S-layers) form symmetrical lattices on many bacterial and archaeal cells, playing essential roles such as facilitating cell adhesion, evading the immune system, and protecting against environmental stress. However, the experimental structural characterization of these S-layers is challenging due to their self-assembly properties and high sequence variability. In this study, we introduce the SymProFold pipeline, which utilizes the high accuracy of AlphaFold-Multimer predictions to derive symmetrical assemblies from protein sequences, specifically focusing on two-dimensional S-layer arrays and spherical viral capsids. The pipeline tests all known symmetry operations observed in these systems (p1, p2, p3, p4, and p6) and identifies the most likely symmetry for the assembly. The predicted models were validated using available experimental data at the cellular level, and additional crystal structures were obtained to confirm the symmetry and interfaces of several SymProFold assemblies. Overall, the SymProFold pipeline enables the determination of symmetric protein assemblies linked to critical functions, thereby opening possibilities for exploring functionalities and designing targeted applications in diverse fields such as nanotechnology, biotechnology, medicine, and materials and environmental sciences.

Symmetry patterns are a fundamental and widespread feature, spanning from the microscopic to the macroscopic scale. At the molecular level, the driving force of this symmetry is self-assembly, where individual components autonomously organize into larger, ordered structures driven by their interactions. Many proteins have symmetrical structures, which are essential for their function[1]. Examples of this symmetrical organization include the hexagonal arrangement of transmembrane chemotaxis receptors, the icosahedral capsid proteins of viruses, and most prominent, the repetitive regular array of S-layer proteins[2–6].

Surface layers (S-layers) are porous 2-dimensional crystalline protein arrays covering the cell envelopes of many eubacterial and archaeal strains[5,6]. The S-layer is built from one or more (glyco)-protein subunits, S-layer proteins (SLPs), that self-assemble into a highly flexible and dynamic lattice through an entropy-driven process, allowing for structural adaptation in response to changing environmental conditions[4,6–11]. So far, S-layers have been shown to serve various functions, including cell stability, adhesion, molecular sieve, and aiding in osmotic stress adaptation[6,12–14]. Still, each S-layer's exact functionality and assembly properties remain unclear. Additionally,

[1]Institute of Molecular Biosciences, University of Graz, Graz, Austria. [2]Medical University of Graz, Graz, Austria. [3]Institute of Nanobiotechnology, University of Natural Resources and Life Sciences Vienna, Vienna, Austria. [4]Structural Biology Unit, Institute of Molecular Biology of Barcelona, Spanish National Research Council, Barcelona, Spain. [5]ICREA, Institució Catalana de Recerca i Estudis Avançats, Barcelona, Spain. [6]Field of Excellence BioHealth, University of Graz, Graz, Austria. [7]BioTechMed-Graz, University of Graz, Graz, Austria. [8]These authors contributed equally: Christoph Buhlheller, Theo Sagmeister. ✉e-mail: tea.pavkov@uni-graz.at

S-layers represent a distinct structural basis for generating complex supramolecular assemblies with considerable application potential in (nano)biotechnology, biomimetics, biomedicine, and synthetic biology[6,15]. The ability to self-assemble also plays a crucial role for viral capsid proteins, which form envelopes encapsulating the viral nucleic acid[16]. Successful assembly and disassembly of the virus proteinaceous coat is crucial in virus replication and infection[17].

Classical structure determination methods are often insufficient for obtaining atomic resolution insights into fully assembled S-layers due to their supramolecular assembly properties. We recently demonstrated that a fully assembled model from individual domain fragments can be obtained for the SlpA S-layer protein of *Lactobacillus acidophilus*[18]. However, determining the structure of fully assembled S-layers remains challenging, and so far, atomic-level assemblies have been solved for only a few species (Supplementary Table 1). The fast-advancing development and accuracy in structure prediction programs like RoseTTAFold[19], AlphaFold2[20], and AlphaFold-Multimer[21] are bridging the gap between missing experimental structures and protein interactions. Here, we present Symmetry Protein Fold (SymProFold), a pipeline for predicting the fully assembled proteins with certain symmetry and unit cell parameters, using only sequential information as input, without prior knowledge of symmetry or oligomerization state, compared to other methods.

AlphaFold has been extensively used to predict individual virus proteins[22,23] and a method to combine AlphaFold monomer predictions with symmetric all-atom docking simulations to predict cubic complexes is available[24]. Several methods are currently available for the prediction of large oligomeric complexes which require besides the sequence additional information as input to ensure a reliable output. Depending on the method used in addition to the protein sequence, either stoichiometry information[25-27] or symmetry group information[28] are needed. Schweke et al. present a pipeline for the calculation of cyclic homo-oligomers using sequential information and

the program AnAnaS[29-31] for the identification of symmetries, concentrating on cyclic symmetries and dihedral or cubic groups.

In this work, we present SymProFold a method specifically designed for the prediction of supramolecular 2D assemblies, as found in S-layers and virus capsids. The assemblies of 19 S-layers from bacteria and archaea as well as a viral capsid are predicted, which were not part of the AlphaFold-Multimer training set and whose structure was hitherto not solved (Supplementary Table 2). The generated SymProFold models were validated using experimental data both at the cellular and atomic levels, when available. The predictions provide comprehensive insight into the scarcely understood assembly of S-layers at an atomic level and reveal exciting features for S-layers which, considering the central function of S-layers in microorganisms, will be crucial for clarifying S-layer functionality and enabling the design of S-layers for their usage in a broad range of applications.

## Results

### Workflow of SymProFold

The underlying idea behind SymProFold is to combine oligomer predictions with the general symmetry patterns found in nature, as observed for S-layers, for generating a model of fully assembled layers. An overview of the general workflow is shown in Fig. 1 and is described in detail in the Methods section.

SymProFold uses the sequential information of the protein of interest, therefore, no prior knowledge of the symmetry or size of the unit cell of the assembly is needed. The protein analysis process begins with the definition of protein domains, which are defined either manually or via the tool 'Domain_Separator' (Supplementary Method 1). Subsequently, full-length and truncated variants of the protein, called subchains, are created according to specified domains (Supplementary Method 2, Supplementary Table 3). These subchains are then used for oligomer predictions, forming different symmetric complexes (dimers to hexamers), which are evaluated for rotational

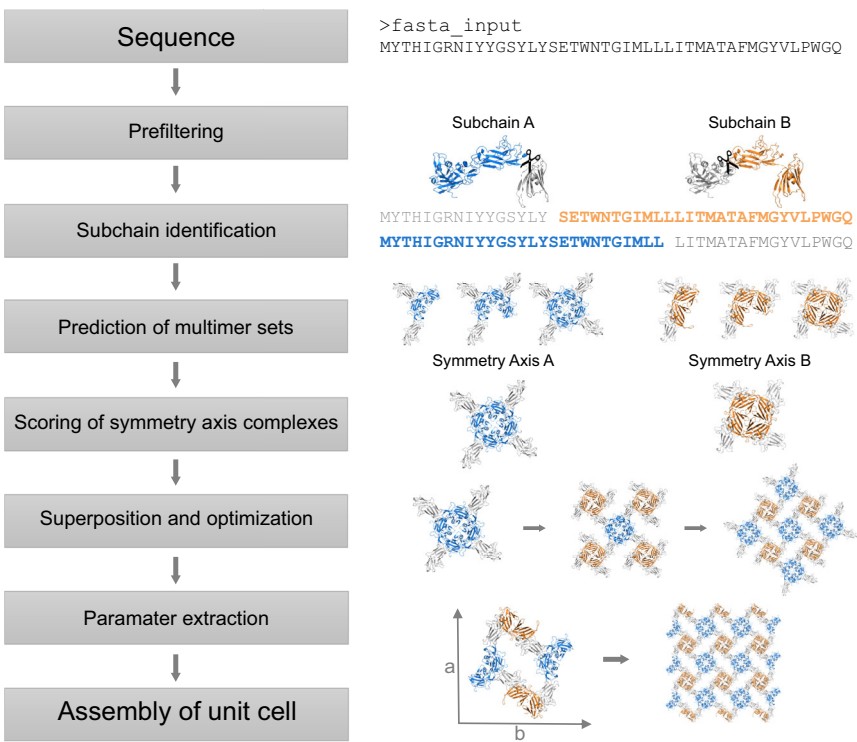

**Fig. 1 | General workflow of SymProFold.** The symmetry and unit cell are calculated starting from the sequence and utilizing AlphaFold-Multimer for oligomer predictions. An example of *A. salmonicida* is shown, with subchain 12 in blue and subchain 23 in orange. The final output is the minimal description of the unit cell parameters.

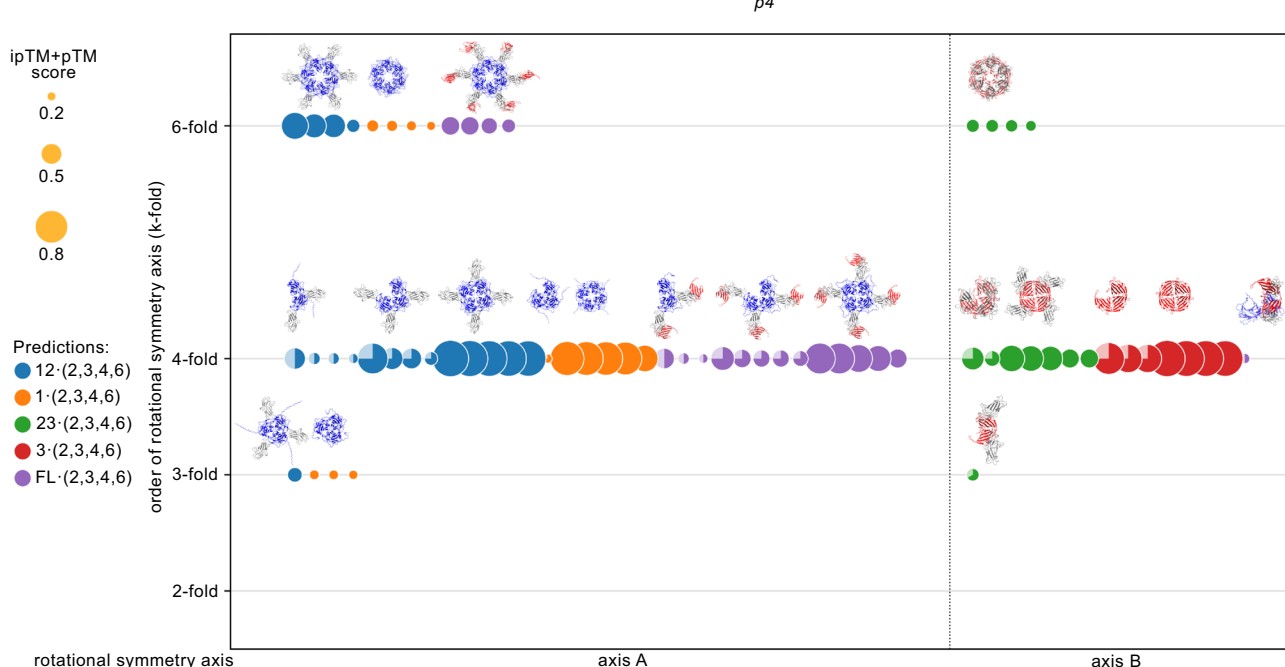

**Fig. 2 | Clustering of predicted symmetry complexes for a *p4* S-layer from *A. salmonicida*.** The circle diameters represent the respective score of each prediction (exact scores are listed in Supplementary Fig. 6); each color represents a subchain (blue: 1–2, orange: 1, green: 2–3, red: 3, violet: full-length). The symmetry axis order (k-fold) determined by prediction is depicted on the y-axis. Up to five symmetry complexes with the highest score are shown for each subchain.

Predictions with scores below 0.2 are not shown. Symmetry complexes are also clustered according to their binding interfaces (x-axis). Dimer and trimer predictions can also result in models that satisfy a 4-fold rotational axis. This is indicated with a ½ or a ¾ filled circle at the 4-fold axis respectively. The structure of each predicted multimer set is shown.

symmetry. The need for using subchains arises from three main reasons. Large systems (>3000–4000 amino acids) exceed the computing power of the used hardware, one symmetry center is strongly favored and hinders reliable prediction of the second symmetry center, or an assembly is impossible due to twisted arrangements of full-length models (Supplementary Table 4). The symmetry complexes are then filtered based on (i) rotational symmetry, (ii) a weighted model confidence score (0.8*ipTM+0.2*pTM, further referred to as ipTM+pTM score) of at least 0.2, (iii) the number of clashes, and (iv) no unusually high fraction of intermolecular β-strands (Supplementary Method 3). An example output of the filtering and clustering of symmetry complexes from *A. salmonicida* S-layer exhibiting a *p4* symmetry is shown in Fig. 2. Prediction scores suggest the presence of two rotational symmetry axes (A, B), each exhibiting a 4-fold symmetry. A complete overview of our proposed models' top-scored symmetry complexes and clustering is recapped in Supplementary Fig. 2 and all individual plots for each prediction are shown in Supplementary Figs. 3–20. Compared to full-length predictions, subchain predictions can result in higher ipTM+pTM scores due to the omission of domains connected via flexible linkers and their introduced uncertainty reflected in lower scores.

**Assessment of prediction quality**
Eventually, the symmetry complexes are clustered by their binding interfaces, and potential repetitive assemblies are tested by superposition and scored with the aim of identifying the most probable representation of a fully assembled S-layer. For SLPs listed in Table 1, high-scoring SymProFold S-layer models were obtained (Fig. 3), corresponding well with the experimentally determined S-layer parameters. The structures shown are representative of the calculated ensemble of SymProFold models exhibiting high-quality scores (see "Methods – Parameter extraction"). The primitive unit cell of each

S-layer is shown in Supplementary Fig. 29. In general, the symmetry complex which forms the rotational symmetry axis A had a slightly higher median ipTM+pTM score of 0.80 than axis B with a median ipTM+pTM of 0.74. Both axes combined result in a medium ipTM+pTM score of 0.77, indicating good prediction results (Supplementary Method 6, Supplementary Fig. 22, Supplementary Table 5). The number of effective sequences (Neff)[32] in the MSA for the individual subchains was calculated using NEFFy[33], which did not correlate with a successful outcome. Subchains with an integrated Neff of below 5 still can result in an assembled layer (Supplementary Table 7). For *p1* S-layers of EA1 from *Bacillus anthracis* and the main SLP from *Bacillus licheniformis*, an augmented set of single-domain subchains and heterodimer predictions with the full-length protein were used as described in the methods section and Supplementary Method 7 (Supplementary Figs. 23, 24).

**Validation of models with experimental data**
We selected a broad range of S-layer proteins from Gram-positive, Gram-negative bacteria and archaea and validated calculated assembly models with experimentally published data (Table 1, Supplementary Table 1–2).

The unit cell parameters extracted by SymProFold of the assembled models correlate with literature values (Table 1). If available, we also compared the experimental microscopy data with our models (Fig. 4). Overall, the domain arrangement and assembly of our predictions agree well with the published data, showing average differences of 5% regarding cell parameters (Table 1). The SymProFold predicted S-layer assemblies reveal detailed structural insights where experimental studies provide limited and ambiguous information. For S-layers from *Vibrio aerogenes, Paenibacillus naphtalenovorans, Pyrococcus abyssi, Methanococcus voltae, Thermococcus camini, Thermococcus thioreducens,* and *Phocaeicola vulgatus*, no experimental data

**Table 1 | Comparison of lattice constants determined by prediction and experimentally**

| Organism | Accession No | Amino acids | Sym. | Unit cell prediction [nm] | Unit cell experimental [nm] | Experimental Data |
|---|---|---|---|---|---|---|
| *Bacillus anthracis* | P94217 | 862 | *p1* | a: 7.3<br>b: 8.9<br>γ: 105° | a: 6.9<br>b: 8.3<br>γ: 106°[68] | Neg. stain projection map |
| *Desulfurococcus mucosus* | E8R795 | 904 | *p4* | 17.5 | 18.0[69] | Neg. stain tilt series |
| *Aeromonas salmonicida* | P35823 | 502 | *p4* | 12.5 | 12.1[70]<br>12.5[71] | Neg. stain tilt series |
| *Sporosarcina ureae* | Q0VJW4 | 1097 | *p4* | 14.0 | 12.9[72] | Neg. stain tilt series |
| *Aneurinibacillus thermoaerophilus* | Q6TL21 | 738 | *p4* | 10.6 | 10[73] | Freeze etch |
| *Corynebacterium glutamicum* | Q2VRQ3 | 491 | *p6* | 17.1 | 16.0[74] | AFM |
| *Methanococcus vannielii* | A6URZ5 | 567 | *p6* | 10.8 | 12.0[75] | Crystal structure<br>*M. acetivorans* SLP |
| *Brevibacillus brevis* | P06546 | 1053 | *p6* | 17.0 | 17.8[76] | Neg. stain projection map |
| *Thermoanaerobacter kivui* | P22258 | 762 | *p6* | 18.1 | 19[77] | Neg. stain tilt series |
| *Viridibacillus arvi* | A0A0K2Z0V7 | 1016 | *p4* | 13.2 | 13.3[72] | AFM, crystal structure presented in this study* |
| *Methanococcus voltae* | Q50833 | 576 | *p6* | 10.1 | | crystal structure presented in this study** |
| *Bacillus licheniformis* | P49052 | 874 | *p1* | a: 7.5<br>b: 9.3<br>γ: 82° | | No experimental data |
| *Vibrio aerogenes* | A0A1M5ZCF8 | 646 | *p4* | 12.7 | | No experimental data |
| *Vibrio quintilis* | A0A1M7YYM3 | 642 | *p4* | 12.2 | | No experimental data |
| *Paenibacillus naphtalenovorans* | A0A0U2M877 | 1053 | *p4* | 12.4 | | No experimental data |
| *Pyrococcus abyssi* | Q9V0N3 | 604 | *p6* | 18.2 | | No experimental data |
| *Thermococcus camini* | A0A7G2D8K1 | 489 | *p6* | 14.4 | | No experimental data |
| *Thermococcus thioreducens* | A0A0Q2M111 | 607 | *p6* | 18.5 | | No experimental data |
| *Phocaeicola vulgatus* | A0A5P3AY64 | 1100 | *p6* | 25.9 | | No experimental data |

*PDB 9FS9;
**PDB 9FSA

on the unit cell parameters, structure, or symmetry are reported. Still, SymProFold predictions of the mentioned S-layers exhibit high output scores, indicating the possible symmetry and structural architecture of these S-layers (Fig. 3).

High-resolution experimental data confirming the proposed assemblies at the atomic level is very scarce. Based on our predicted models of *Viridibacillus arvi* and *Methanococcus voltae*, we designed constructs (see Methods section) containing only the domain responsible for the formation of either the 4-fold or 2-fold axis. We successfully obtained crystal structures for both (PDB 9FS9; PDB 9FSA) and compared them to the predicted models (Fig. 4G, H). Aligning the crystal structures with the SymProFold models revealed a great similarity, with RMSD values of 0.65 Å for *V. arvi* and 1.38 Å for *M. voltae*, further confirming our predicted assemblies. Recently, high-resolution structures for EA1 of *Bacillus anthracis*[34] and the SLP from *Nitrosopumilus maritimus*[35] (not included in our initial dataset) were reported. Both structures agree well compared to our SymProFold models (Supplementary Figs. 25–27).

### Exploring distinct features of predicted S-layer assemblies

To date, the structures of only a few SLPs have been reported (Supplementary Table 1). The assembly data acquired for S-layers whose structure was not yet known provides a foundation for the functional analysis of distinct S-layers, as well as for delineating distinct features within this protein family (Supplementary Fig. 28). As observed for *D. mucosus*, pore sizes can be enormous (Supplementary Fig. 28I), indicating that in this individual case, the flexibility of the layer is more critical than the barrier function. Most of the other predicted S-layer assemblies form a tight network, acting as an efficient filter selectively

allowing molecules to pass in and out. Moreover, depending on the environmental conditions[36–38] either flexibility or mechanical stability of the S-layer is of greater importance. The length of the structural elements linking the surface exposed region to the membrane provides additional valuable information about the thickness of the periplasmic-like space in bacteria and archaea (Supplementary Fig. 28I).

The prediction of the *C. glutamicum* S-layer reveals a single domain α-helical protein with *p6* symmetry (Supplementary Fig. 28E). The C-terminus ends in a predicted transmembrane helix resembling the anchoring mechanism of archaeal S-layers (Supplementary Fig. 28F,G). The SymProFold prediction of S-layers from *P. abyssi, M. voltae, T. camini, T. thioreducens*, and *M. vannielii*, even though different in sequence, size, and domain structure, show an analog domain arrangement as the crystal structure of *M. acetivorans* SLP (Supplementary Fig. 27) and reveal a dimeric anchor, which has not been described previously (Supplementary Fig. 28A–D). For the assembly prediction of S-layers from *B. brevis* and *V. arvi* (Slp1), an additional region above the core, responsible for the self-assembly, is present. The domains and residues in this layer might be important for interaction with the environment, including the host. In addition, the fully assembled S-layer gives valuable information about the orientation of the layer and surface properties as the electrostatic potential, revealing potential functions (Supplementary Fig. 30).

### Prediction of viral capsids

In the case of viral capsids, a curved surface favors a 5-fold rotational symmetry (Supplementary Fig. 21). SymProFold predicted an icosahedral viral capsid from *Odonata-associated circular virus 21* (T = 1)

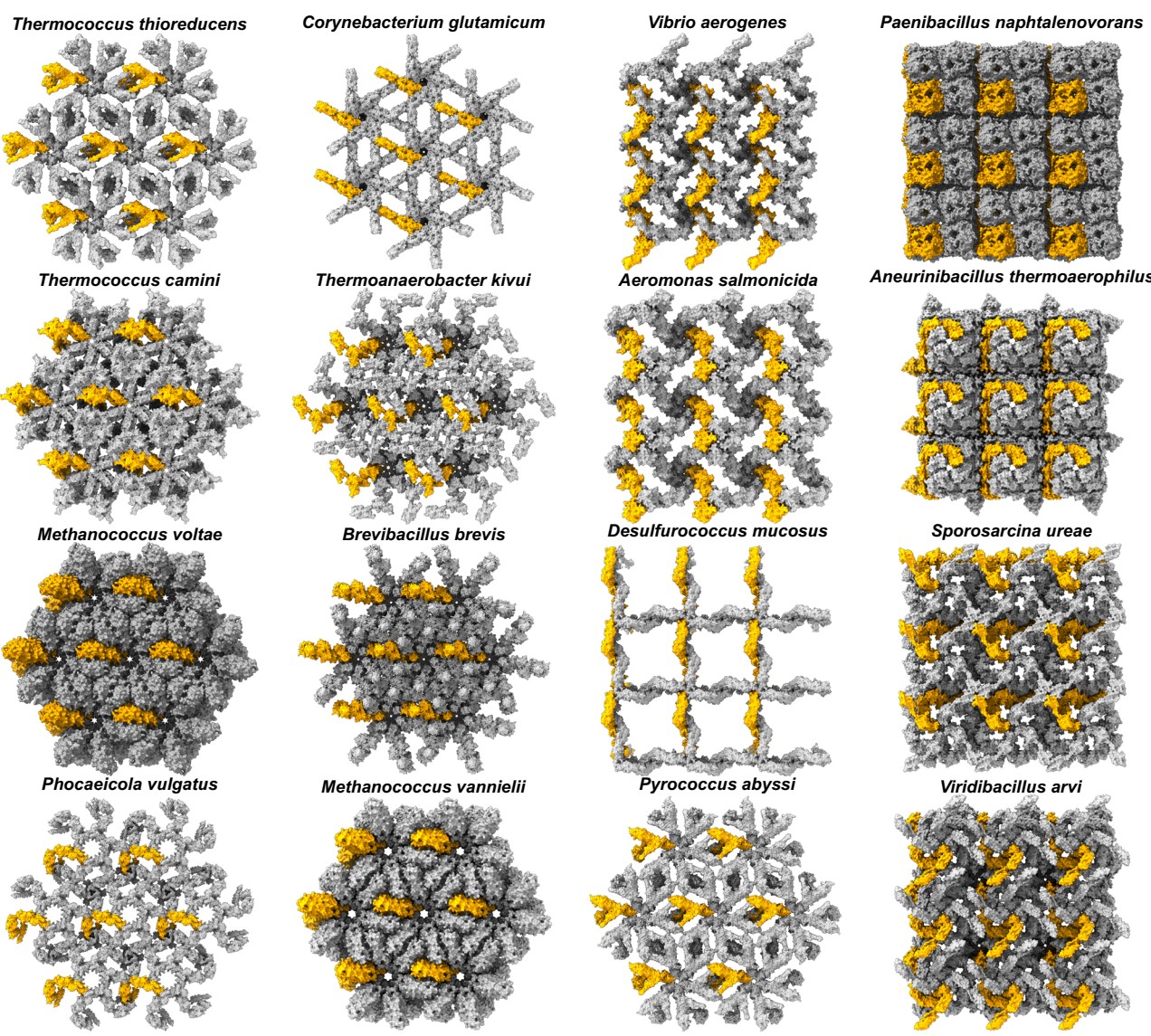

**Fig. 3 | Top view of the calculated S-layers.** Successfully calculated assemblies of different S-layers are shown in surface representation from the top side (outside of the cell, facing the environment). One monomer of the assembly unit is shown in orange.

belonging to the *Smacoviridae*[39]. Analog to the regular SymProFold, two symmetry complexes were calculated and superimposed to one fully assembled tile (Fig. 5A–C). The curvature is present in the calculated tile (Fig. 5D, E) that, when manually superimposed on each other, generates the full viral capsid (Fig. 5F). This model would have a diameter of ca. 23 nm. There is no experimental data for the diameter of *Smacoviridae*, but close members of the same phylum *Cressdnaviricota* show a diameter of 17–20 nm (*Nanoviridae*) and 20–22 nm (*Genomviridae*)[40], which correlates with our proposed model. Compared to the median ipTM+pTM score (Supplementary Fig. 22) of our benchmark cases (axis A: 0.80, axis B: 0.74), the virus example shows slightly lower ipTM+pTM values for axis A 0.72 and axis B 0.55, but are still within a typical range of the predicted models.

**Prediction of symmetrical oligomers with only one symmetry axis**
SymProFold was further tested on proteins that do not form symmetrical 2D arrays yet are known, through literature or existing crystal structure data, to possess a rotational symmetry axis. The circadian clock protein KaiC (Uniprot: Q8GGL1) forms symmetrical ring-like shaped hexamers[41]. SymProFold can reliably predict the

6-fold axis with high model confidence scores. Another 2-fold axis was found with extremely low scores just above the cutoff value of ipTM+pTM of 0.2 (Supplementary Fig. 31). At the superposition step, it was not possible to generate an assembly in a plane without obtaining severe clashes. Therefore, it was automatically filtered out. The crystal structure of YabJ (PDB 5Y6U[42]) forms a homotrimer (Supplementary Fig. 32A). SymProFold identified a 3-fold symmetry complex with a high ipTM+pTM of 0.96, which is higher than the median score of the benchmark cases probably due to the structure's association in the AlphaFold-Multimer training set (Supplementary Fig. 33). A 4-fold symmetry complex was predicted with much lower scores, showing the same main binding interfaces as the 3-fold prediction, and therefore was clustered to the same axis. No additional symmetry axis was predicted, therefore a superposition of two different axes is not possible. As a third example, we tested the sequence of N9 neuraminidase from the *Influenza A virus*. The crystal structure (PDB 6MCX[43]) shows a symmetrical 4-fold axis (Supplementary Fig. 32B). For this protein, SymProFold predicts a single cluster with a 4-fold rotational symmetry axis (Supplementary Fig. 34). Since there is only one axis of rotational symmetry, spanning a 2D layer is not possible. For all of our three presented test cases,

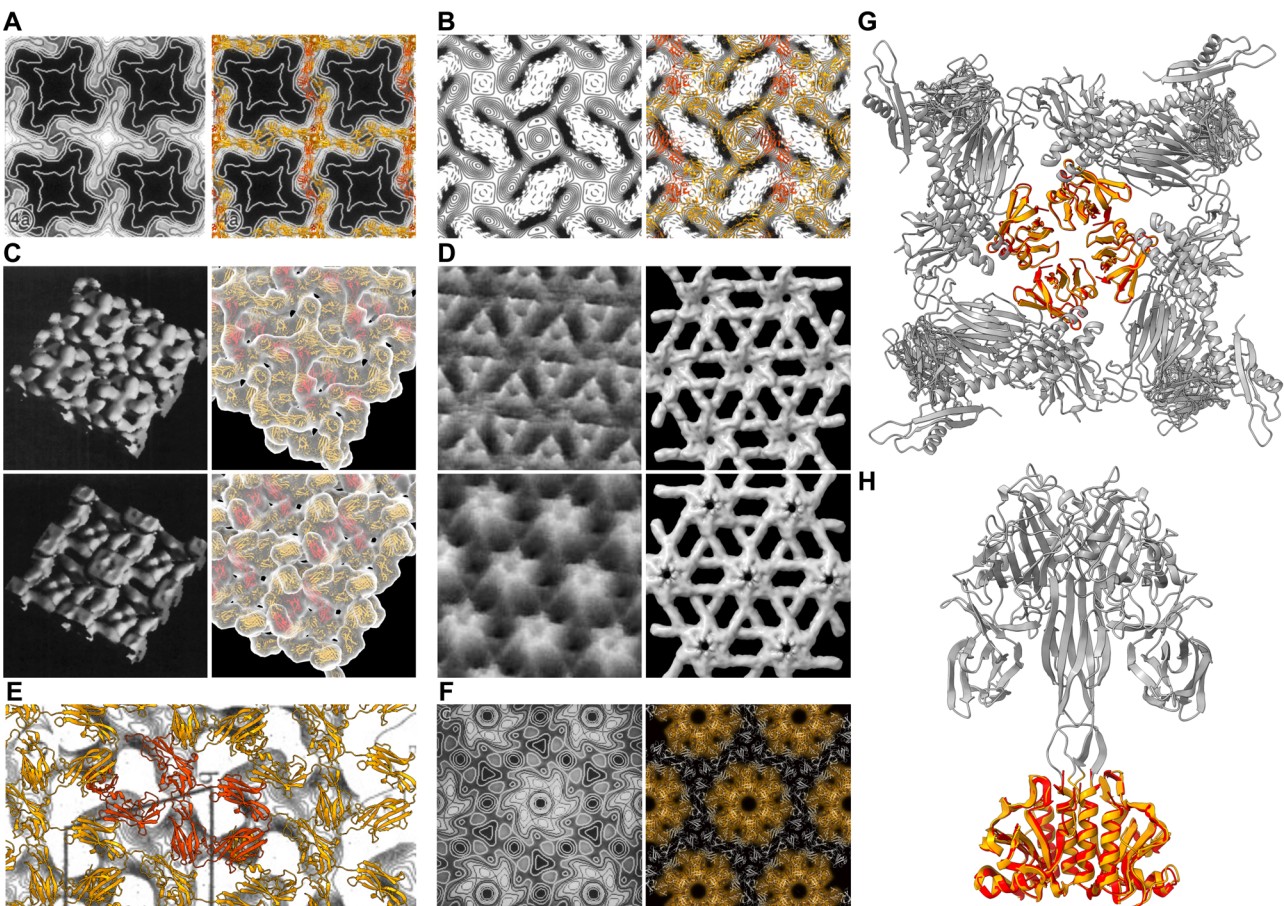

**Fig. 4 | Comparison of experimental data with predicted assemblies.** Electron microscopy data obtained from a negative stain tilt series for **A** *D. mucosus* S-layer (adapted from Wildhaber 1987[69]) and **B** *A. salmonicida* S-layer (adapted from Dooley 1989[70]) compared to the SymProFold prediction (in ribbon representation). **C** 3D representation of the outer (top) surface and inner (bottom) surface of *S. ureae* S-layer as reported by Engelhard 1986[72] compared to the calculated density representation from the predicted assembly with ChimeraX molmap. **D** High-resolution AFM topographs (adapted from Scheuring 2002[74]) of the outer (top) surface of *C. glutamicum* S-layer showing triangular-like features around the 3-fold axis and flower-like features around the 6-fold axis on the inner surface (bottom). The same is observed in the calculated structure shown as a calculated density map (molmap in ChimeraX). **E** Electron microscopy obtained projection map (adapted from Couture-Tosi 2002[78]) of the *p1* symmetry protein EA1 from *B. anthracis* compared to the calculated assembly. **F** Reconstructed averaged micrographs from *B. brevis* S-layer, adapted from Tsuboi et al. 1989[76] and superimposed SymProFold prediction. The 6-fold axis shows the same counterclockwise wheel-like symmetry. **G** Proposed assembly of *Viridibacillus arvi* (gray) with the 4-fold symmetry axis shown in orange. The crystal structure (PDB 9FS9 of the domain responsible for the four-fold axis (amino acids 765–844) is aligned (red). **H** Proposed assembly of *Methanococcus voltae* (gray) with the 2-fold symmetry axis of the anchoring region is shown in orange. The crystal structure (PDB 9FSA) of the domain responsible for the dimerization of the anchor (amino

acids 24–75 and 484–576) is aligned (red). Figure **A** used with permission of American Society for Microbiology · Journals, from "Three-dimensional structure of the surface protein of *Desulfurococcus mobilis*"; Wildhaber, I., Santarius, U. & Baumeister, W., J Bacteriol 169, 5563–5568 (1987); Figure **B** used with permission of American Society for Microbiology · Journals from "Three-dimensional structure of an open form of the surface layer from the fish pathogen *Aeromonas salmonicida*"; Dooley, J. S., Engelhardt, H., Baumeister, W., Kay, W. W. & Trust, T. J., J Bacteriol 171, 190–197 (1989); Figure **C** used with permission of American Society for Microbiology · Journals, from "Three-dimensional structure of the tetragonal surface layer of *Sporosarcina ureae*"; Engelhardt, H., Saxton, W. O. & Baumeister, W., J Bacteriol 168, 309–317 (1986); Figure **D** used with permission of John Wiley & Sons - Books, from "Charting and unzipping the surface layer of *Corynebacterium glutamicum* with the atomic force microscope"; Scheuring, S. et al., Mol Microbiol 44, 675–684 (2002); Figure **E** used with permission of American Society for Microbiology · Journals, from "Structural Analysis and Evidence for Dynamic Emergence of *Bacillus anthracis* S-Layer Networks"; Couture-Tosi, E. et al., J Bacteriol 184, 6448–6456 (2002); Figure **F** used with permission of American Society for Microbiology · Journals, from "In vitro reconstitution of a hexagonal array with a surface layer protein synthesized by *Bacillus subtilis* harboring the surface layer protein gene from *Bacillus brevis* 47"; Tsuboi, A. et al., J Bacteriol 171, 6747–6752 (1989); All permissions for figures **A**–**F** conveyed through Copyright Clearance Center, Inc.

SymProFold stopped and no 2D layer was falsely positively generated for proteins with just one rotational axis.

## Discussion

SymProFold can predict the structural organization of higher symmetrical assemblies, such as S-layers, as present in their native state in assembled form at the cell surfaces. As only sequential information is required as an input, this approach enables numerous avenues for investigating symmetrical formations, even those that pose significant experimental challenges, such as high toxicity, pathogenicity, extreme

growth conditions, and substantial costs for the experimental investigation. Furthermore, the self-assembly property and significant sequence variation of S-layers present a challenge for structural characterization using techniques such as X-ray crystallography and electron microscopy.

Due to the considerable size diversity of SLPs, some exceed 1000 amino acids[6], predictions of tetramers or hexamers can easily exceed the computing power needed to get highly confident models with AlphaFold. Nevertheless, the rapid improvements of AlphaFold in predicting large protein complexes will increase the size limit of

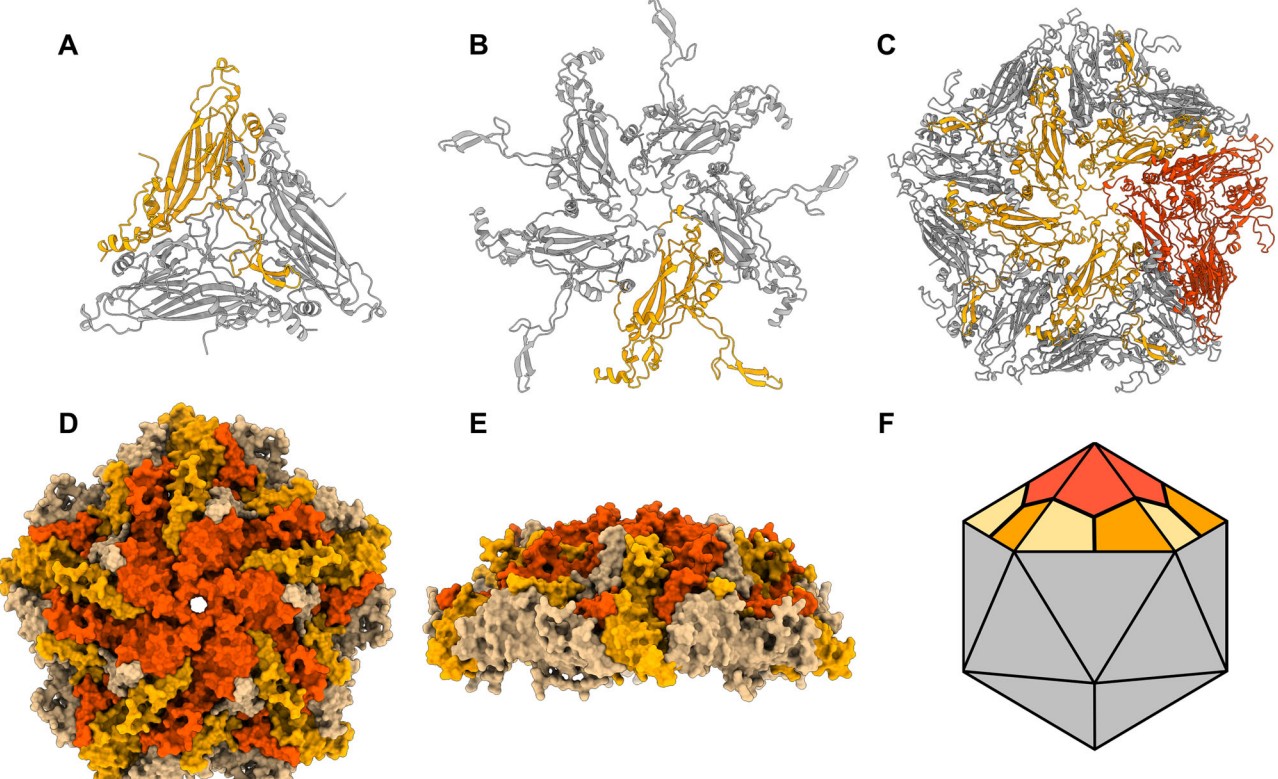

**Fig. 5 | Prediction of a viral capsid of *Odonata-associate circular virus 21*.** The prediction of the *Odonata-associate circular virus 21*(Uniprot:A0A0B4UH63 [https://www.uniprot.org/uniprotkb/A0A0B4UH63/entry]). **A** 3-fold and **B** 5-fold subunit. One monomer is colored in orange. **C** Superimposed and assembled to the full assembly unit, the 5-fold axis is shown in orange, and the 3-fold axis is shown in red. In **D** and **E**, the assembled model is shown in surface representation from the top and side views. The three chains of the 3-fold unit are colored red, orange, and beige. **F** shows the icosahedral (T = 1) assembly symmetry of the viral capsid. Colored according to the tiles shown in **D**, **E**.

SymProFold calculations. Calculated models can potentially be improved by manual pre-processing of input sequences according to prior knowledge. Regions like signal sequences or domains not involved in the assembly, such as cell wall anchors, can be removed before the calculation, thereby reducing the size of the input sequence. The SymProFold method has several restrictions that lead to early termination of the pipeline when sequences of non-assembling proteins are used. The SymProFold pipeline terminates if at least one of the following cases is true: the axis tilt is above 45° (SymProFold supports 2-, 3-, 4- or 6-fold rotational symmetry), the gap between the predicted symmetry complexes is too large (and they cannot be connected), the deviation between the calculated lattice constant (unit cell) and the model (by superposition) is too large, or a symmetry axis complex does not have a rigid folding unit.

In case SymProFold fails to detect more than one strong intermolecular interaction, the construction of a fully assembled layer is not possible and results in a partially assembled model. This limitation causes a loss of information regarding weak but critical interactions, which may be required for a complete assembly. S-layers and viral capsids, which consist of multiple proteins, also represent a challenge for SymProFold predictions. In such cases, prior knowledge may be needed as the identification of the interactions important for the initiation of the assembly could be difficult via the automatic pipeline. Nevertheless, manual intervention and/or an extension of the software could overcome this limitation and enable the prediction of assemblies composed of more than one protein. Further adaptations in SymProFold are also needed for viral capsid automated predictions to address potential challenges like pseudosymmetry, higher triangulation number, and variations in assembly curvature.

Many biological functions of S-layers depend on the completeness of the cell coverage as well as the structural and physicochemical repetitive uniformity, down to the subnanometer scale. Obtained structural data of assembled S-layers reveal properties of the exposed surface regions (Supplementary Fig. 29), internal pores, and anchoring domains essential for cell attachment and highlight interaction interfaces within the layer. Understanding these aspects is crucial for elucidation of the specific function of a particular S-layer, depending on the microorganism. This is especially important in pathogenic bacteria since S-layers play a role in surface adhesion and in interactions with the host's immune system. Furthermore, numerous studies on the in vivo and in vitro morphogenesis of S-layers demonstrated that lattice growth on growing cells is a highly dynamic process[6,7,11,12,44–47]. Approximately 500 subunits per second must be synthesized at high growth rates, translocated to the cell surface, and incorporated in a defined orientation to the existing lattice while maintaining an equilibrium of the lowest free energy[6,46–49]. The adaptable lattice of S-layers on growing and dividing bacterial and archaeal cells represents an advanced evolutionary stage in morphogenesis[46]. At these specific sites, bonds must swiftly open and re-form. These dynamics are also present in the assembly of the capsid during the viral life cycle, including disassembly upon infection and reassembly during viral packaging. To gain insights into these dynamic processes, the SymProFold predicted assemblies can serve as a starting point to identify the required interactions within the lattice, perform molecular dynamic studies, and investigate binding events with ligands or receptors. S-layers often represent the outermost layer, therefore environmental conditions could impact their structure and functionality. Using the predicted assemblies in combination with

computational methods, the influence of e.g. pH changes on the S-layer could be investigated.

Structural insights into the S-layer assembly offer a vast opportunity for developing technologies that can mimic the distinct properties of these versatile and adaptable structures. Rational engineering of artificial structures with different properties could include the design of materials with advanced self-assembly properties or the creating improved drug delivery systems and biosensors[50].

Since the S-layers of pathogens serve as favorable regions for drug targeting[36,51], understanding of anchoring mechanisms and/or interactions within the self-assembled S-layer provides the ideal basis for rational drug design and enables the development of strategies for weakening S-layer protective function. Furthermore, structural information now allows for analyzing the antifouling properties of S-layer lattices[6] and offers a foundation for mimicking these structures in polymer technologies[52]. The precise biophysical characterization of the pores allows for custom alterations of S-layer permeability, which was previously possible only through chemical modifications[12]. S-layers have also shown great potential as a platform for drug delivery because of their biocompatibility, stability, and regular pore structure. Lipidic nanoformulations such as liposomes, solid lipid nanoparticles (SLNs), or emulsomes as drug delivery systems show better resistance to oxidative stress and membrane damage when covered with a crystalline S-layer[53,54]. Uptake studies with emulsomes coated with S-layer proteins did not show significant cytotoxicity by human liver carcinoma cells (HepG2). The capacity to recrystallize S-layer proteins in lipidic formulations allows extremely precise targeted delivery of specific antibodies in high concentrations[55,56] or drug-loaded particles, especially poorly water-soluble targets such as antimicrobial peptides or easily degradable biologicals such as enzymes used for the enzyme replacement therapy[6,53–55,57]. Not yet resolved functionalities, such as cellular targeting and generation of fusion proteins with specific properties[58] can be introduced. The S-layers of bacteria and archaea in the human microbiome are perfect candidates for these biotechnological applications. Altogether, this opens a promising chapter in rational engineering and adaptation of many S-layers for applications in medicine and diagnostics, like treating fungal and viral infections, dermal conditions, cancer, immune deficiency, or rare genetic disorders.

## Methods

### Pre-filtering of possible prediction candidates

SymProFold uses sequence files in fasta format as input and consequently checks their predictability by performing homodimer predictions with AlphaFold-Multimer[21]. Homodimer predictions are scored according to the ipTM+pTM score, a weighted model confidence score for complex predictions, which ranges from 0–1 and is calculated using 80% of the ipTM score and 20% of the pTM score as described by Evans et al. 2022[21]. Only homodimer predictions with a minimal score of 0.3 ipTM+pTM are further processed, excluding proteins that are unlikely to dimerize. The sequences of 19 annotated SLPs, 3 non-SLPs, and viral capsid from *Odonata-associated circular virus 21* were used as input. Table 1 includes a list of proteins for which SymProFold S-layer predictions were performed. We used the annotated S-layer protein for each species, as reported in the Uniprot databank, for all of the presented S-layer predictions and calculations.

### Identification of domains and subchains

As a next step, domains of the protein need to be defined in a fasta file with domains separated by line breaks. Domain boundaries can be set manually, or an automated domain identification can be used. Both start with predicting the full-length protein structure as a monomer using AlphaFold. For automated domain identification, we created a 'Domain_Separator' tool that can be included in the SymProFold pipeline to prepare the fasta file. Domain_Separator is a Python script

that uses a coordinate file as input, identifies structural domains, and creates a fasta file with domain sequences separated by line breaks. Additionally, a ChimeraX file with separately colored domains can be generated. In an iterative process, small domain subsections are merged until they describe a complete domain. The initial domain subsections are chain ranges created through local crosslinking of secondary structures. Using relations between contact areas, surface areas, and subsection sequence lengths, neighboring domain subsections are then iteratively merged into even larger domain subsections. This iterative process ends when a saturation of merging is reached. In a postprocessing step, for each linker between domain sections, a cropping point is identified that minimizes the contacts between both domains. An example of Domain_Separator utilization is found in the Supplementary Method 1 and Supplementary Fig. 1. The generated fasta file can be used as direct input for SymProFold. Both methods lead to comparable results for identifying domains in our presented examples.

Once the domains are defined in the fasta file, a set of five different subchains of the protein is generated. These subchains represent truncations of the protein according to a scheme described in Supplementary Table 3, which includes the full-length sequence, a subchain without the N-terminus, a subchain without the C-terminus, the first third of the domains, and the last third of the domains. Minimum or maximum lengths were defined (Supplementary Table 3) to mitigate the effects of very large or small domains.

### Prediction of oligomer sets and filtering

For each subchain, the algorithm starts different oligomer predictions (dimers, trimers, tetramers, and hexamers). Five models are calculated for each prediction, and the resulting symmetry complexes are evaluated and further processed.

All predictions are evaluated to check if there is a rotational symmetry axis that can align all the subchain components within the oligomer prediction to each other, satisfying the symmetry requirement. Therefore, the algorithm checks whether the occurring angles between the monomers correspond to a 2-, 3-, 4- or 6-fold rotational symmetry and whether the associated rotational symmetry axis is uniform within a tolerance range (max. of 5 Å deviation for each monomer). Models with an ipTM+pTM score of ≥0.20 and existing interfaces between the monomers are treated as symmetry complexes. The order of the rotational symmetry axis (k-fold) of each complex is deduced from its symmetry angle (2-, 3-, 4-, or 6-fold rotational symmetry axis). For example, a trimer prediction of a given subchain of an SLP that forms a *p6* S-layer could result in a 3-fold axis ($\Delta\varphi = 120°$) or 6-fold axis ($\Delta\varphi = 60°$). In the latter case, the property of a 6-fold symmetry axis is described by only three predicted subunits instead of 6. The symmetry complexes are further filtered by criteria to discard predictions of low quality. With respect to the complete protein chain, relaxed models with at least 3.0 clashes per 100aa (and 60.0 per 100aa for unrelaxed) are excluded. Additionally, in all possible sequence sections of length 200aa, only 6.0 clashes per 100aa for relaxed models (and 120.0 per 100aa for unrelaxed) are allowed. The number of clashes is calculated using the corresponding method in ChimeraX[59]. Models in which a part of the protein chain passes incorrectly through the neighboring molecule are excluded by filtering out models with unusually high fractions of intermolecular β-strands.

### Clustering via interfaces

The symmetry complexes are clustered according to the agreement between their binding interfaces, which are compared via interface matrices (distograms). Each cluster can be viewed as a candidate for a rotational symmetry axis of the assembled S-layer. All symmetry complexes are represented by a node in a network graph, connected by edges. Every edge weight is proportional to a correlation coefficient between the interface matrices of the two symmetry complexes

(nodes) connected by it. A more significant agreement between both interface matrices leads to a larger correlation coefficient (see Supplementary Method 4 for details). This network graph is partitioned using the Louvain[60] method.

Typically, each cluster contains differently cropped subchains, resulting in the same symmetry axis. For each cluster, high ipTM+pTM scores and, where present, several different molecule counts leading to the same order of symmetry are indicators for the fold of the respective rotational symmetry axis.

## Superposition of symmetry complexes

Pairs of symmetry complexes from 2 different clusters (symmetry axes) are tested on the possibility of repetitive 2D assembly formation by the superposition of overlapping regions. At least one domain must constitute the overlapping region. The pairs of symmetry complexes to be tested are selected according to indicators of highest scores in a cluster. In case no clear result is found, more pairs of symmetry complexes with lower ipTM+pTM scores from different clusters can be tested. From a pair of symmetry complexes (A, B), the symmetry complex (A) with the higher order of rotational symmetry is aligned in $z$ direction and then complemented by the corresponding number of copies of the other (B) by overlapping superposition. The symmetry complexes are superposed using the *matchmaker* method of ChimeraX[59]. Subsequently, the rotational symmetry axes of symmetry complexes B are aligned in $z$ direction using linkers between complexes A and B as pivot points. Then the symmetry complexes B are complemented by copies of the symmetry complex A by overlapping superposition and the rotational symmetry axes are aligned likewise.

## Parameter extraction

The superposition of the assembled S-layer is scored based on the average intermolecular clashes per residue and the bending score of the assembly. Assembly bending is assessed by the axial tilt between the rotational symmetry axes of symmetry complexes A and B. The bending score is the RMSD of the angles between the axes of the central symmetry complex A and the nearest neighbor symmetry complex B. The score is normalized so that an angle of 0 rad (0°) corresponds to a bending score of 0, and an angle of $\pi/2$ rad (90°) corresponds to a bending score of 1. Addition of the average clashes per residue (Eq. (1)) and the bending score (Eq. (2)) results in the combined quality score (Eq. (3)).

$$\text{score}_{\text{clash}} = \frac{N_{\text{clashes}}}{N_{\text{res}}} \tag{1}$$

$$\text{score}_{\text{bend}} = \frac{2}{\pi} \sqrt{\frac{1}{n} \sum_{i=1}^{n} (\varphi_i - \varphi_0)^2} \tag{2}$$

$$\text{score}_{\text{quality}} = \text{score}_{\text{clash}} + \text{score}_{\text{bend}} \tag{3}$$

Unit cell parameters are determined using averaged distances calculated from vector differences between the noncentral symmetry complexes A with their central symmetry mate. Function command *clashes* in ChimeraX is used to calculate the clash score (Supplementary Method 3)

## Assembly of S-layer unit cell

A mathematically exact unit cell is created using the determined unit cell parameters. To obtain the best possible model, the quality score can be optimized by variation of the pairs of symmetry complexes. The final output is a mmcif file with the primitive unit cell, the unit cell parameters, and symmetry operations to generate the symmetry mates to get a fully assembled S-layer. Furthermore, in the output folder files from previous steps of SymProFold for a subchain

definition, symmetry complexes (SymPlot) and interface network graphs are available.

## Prediction of *p1* S-layer

If SymProFold terminates during the workflow and cannot create a model of a primitive unit cell, it might be due to a *p1* symmetry of the S-layer. Within a *p1* S-layer, no rotational symmetry axes of order 2 or higher occur, therefore SymProFold needs some adjustments to predict the assembly successfully. If a *p1* symmetry is known or assumed, an alternative SymProFold method for the *p1* S-layer can be manually applied. In the *p1* pipeline, no set of subchains is generated, but a full-length prediction is split either manually or with Domain_Separator into all possible single domains. Sets of heterodimers of the full-length protein and each single domain are calculated. If two or more strong interactions are found, the full-length model is mapped to the individual domains, and the fully assembled layer in both directions is generated. Automated superposition of axis objects can still be applied to generate the fully assembled S-layer.

## Prediction of the viral capsid

As a test case, we selected the viral capsid from *Odonata-associated circular virus 21* (Uniprot: A0A0B4UH63). Allowing for the possibility of a 5-fold rotational symmetry axis in the SymProFold workflow basically allows the prediction of some viral capsids. For this, oligomer predictions are performed for dimers, trimers, tetramers, pentamers, and hexamers. In our test case, the predictions are made for only one subchain. In the filtering step, the oligomer predictions are tested for the presence of rotational symmetry axes that can map subchains in the oligomer prediction to one another and fulfill rotational symmetry operations. Therefore, the algorithm checks whether the occurring angles between the monomers correspond to 2-, 3-, 4-, 5- or 6-fold rotational symmetries and whether the associated rotational symmetry axes are uniform within a tolerance range. Since the surface of viral capsids is not a flat 2D plane, the rotational symmetry axes of symmetry complexes B are not aligned parallel to symmetry complex A. SymProFold currently does not support the automated assembly of viral capsids, but the results from the best scoring rotational symmetry axes can be manually built into an orientation that results in fully assembled capsid.

## Resources

For SymProFold, we present an implementation in Python 3, which uses function libraries of ChimeraX 1.6[59]. The Domain_Separator tool is written in Python and uses ChimeraX function libraries and dssp[61] to determine secondary structures. Contacts between residues are determined using the function *contacts* of ChimeraX and the functions *surface* and *measure* to calculate solvent-excluded surface areas. Domain_Separator uses the Python library NetworkX.

## Computing requirements

An AlphaFold-Multimer 2.3 installation in standard configuration with full databases and v3 model weights was used for all complex calculations. The default value of max. 20 recycling iterations was used. The predictions were calculated at the VSC-5 Vienna Scientific Cluster (Vienna, Austria) on a GPU (NVIDIA A100, 40GB of vram).

## Protein expression

The codon-optimized sequences (*E. coli*) for Varv_VI (*Viridibacillus arvi*; amino acids 765–844) and Mvol_anchor (*Methanococcus voltae*; amino acids 24–75 and 484–576 connected with a GGS linker) were purchased precloned in pET24 (BioCat GmbH, Heidelberg, Germany). The 5 μg of plasmid were dissolved in 50 μl water and 0.5 μl were transformed into One Shot BL21 Star (DE3) Chemically Competent *E. coli* (Thermo Fisher Scientific, Waltham, MA, U.S.A.) and plated on LB agar containing 35 μg/ml kanamycin. Single colonies were picked for overnight

cultures in LB broth containing 35 µg/ml kanamycin. Main cultures in 100 ml LB broth containing kanamycin were inoculated 1/100 with ONC and grown to an OD600 between 0.4 and 0.7 at 37 °C. The expression was induced by the addition of 0.5 mM IPTG, the temperature was reduced to 20 °C and carried out ON. Expression cultures were harvested by centrifugation (2800 g, 20 min, 4 °C). The supernatant was discarded, and the pellet was frozen (−20 °C until further use).

### Protein purification
Expression pellets were thawed and resuspended in 20 ml lysis buffer (50 mM HEPES pH 7.5, 300 mM NaCl) and sonicated for cell disruption (Bandelin Sonoplus sonicator at 80%, 5 cycles, 5 min, on ice). The supernatant containing the soluble protein was filtered (Rotilabo syringe filter, PVDF, pore size 0.45 µm, Carl Roth GmbH and Co., Karlsruhe, Germany) before loading samples onto an ÄKTA pure system from GE Healthcare (Chicago, United states). The HisTrap FF affinity column 5 ml from GE Healthcare was equilibrated with the previously mentioned lysis buffer. The column was washed with 20 column volumes with 5% elution buffer (50 mM HEPES pH 7.5, 300 mM NaCl, 500 mM imidazole). The proteins were eluted with 50% elution buffer, concentrated using Amicon Ultra centrifugal filters (Millipore, Merck KGaA, Darmstadt, Germany) to a volume of 500 µl and subjected to gel filtration with SEC buffer (25 mM HEPES pH 7.5, 150 mM NaCl) with a Superdex 200 Increase 10/300 column (Cytiva, Marlborough, MA, USA). Peak fractions were pooled and concentrated for crystallization.

### Crystallization
Crystallization experiments were performed using vapor diffusion sitting drop in Swissci UVXPO 3 Lens crystallization plates (High Wycombe, United Kingdom). Pipetting was carried out with an Oryx 8 robot (Douglas Instruments, United Kingdom) with 35 µL of condition solution in the reservoir and drops of 0.3 µl protein; the concentration ranges from 10 mg/ml to 22 mg/ml, mixed with 0.3 µl screening solution (JCSG+ eco screen, Molecular Dimensions, Calibre Scientific, Rotherham, UK).

### Data collection and processing
Crystals were frozen in liquid nitrogen and data collection of all crystals was performed at 100 K. Crystal screening and data collection were carried out with an in-house dual port system made up of a MetalJet X-ray source (Excillium, Kista, Sweden), a D8 Venture X-ray diffractometer (Bruker, Billerica, USA) and a Photon III detector (Bruker, Billerica, USA). Data processing was performed with DIALS 3.8[62], and data reduction with pointless and aimless[63]. For the dataset collected for *Methanococcus voltae*, an anisotropic cutoff was applied with the Staraniso webserver[64].

### Structures solution, refinement, and analysis
Both structures were solved by molecular replacement using Phaser[65] with the predictions presented in this manuscript (monomeric fragments of the respective amino acid ranges) as templates. Refinement was performed with Refmac[66] and phenix.refine[67]. Structures were deposited at the RCSB Databank with the PDB codes 9FS9 and 9FSA. Table1 containing the collection and refinement statistics is available in Supplementary Method 8 and Supplementary Table 6.

### Reporting summary
Further information on research design is available in the Nature Portfolio Reporting Summary linked to this article.

## Data availability
The generated crystal structures are available at the RCSB protein database under accession codes 9FS9 and 9FSA. All presented SymProFold models are available under https://github.com/symprofold including a detailed tutorial. Previously published crystal structures mentioned in the paper include: 5Y6U and 6MCX. UniProt entries used in this work are Q2VRQ3, P35823, A0A1M7YYM3, A0A1M5ZCF8, A0A7G2D8K1, A0A5P3AY64, A6URZ5, A0A0Q2M111, A0A0K2Z0V7, P22258, P06546, E8R795, Q6TL21, I3XTG6, Q50833, Q0VJW4, Q9V0N3 and A0A0U2M877. Source Data are provided as a Source Data file. Source data are provided with this paper.

## Code availability
The source code of SymProFold and for Domain_Separator, an installation guide, a tutorial, and the models calculated with SymProFold, are available at the GitHub repository [https://github.com/symprofold]. Source code is also available at [https://doi.org/10.5281/zenodo.13327126].

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

## Acknowledgements

The authors acknowledge the financial support by the University of Graz. Computational results presented have been achieved using the Vienna Scientific Cluster (VSC) (project Nr 71272; T.P.K.). Additionally, funding by the Austrian Science Fund FWF doc.fund Biomolecular Structure and Interactions (grant doi:10.55776/DOC130), Land Steiermark, the City of Graz and Doctoral Academy Graz (BioMolStruct consortium) was received for C.G. and T.P.K. T.S. thanks doc.fund project Molecular Metabolism (grant doi:10.55776/DOC50). N.G. thanks FWF for financial support (grant doi: 10.55776/T1239 and Land Steiermark (Project number: PN 3046). I.U. appreciates support by Ministry of Science and Innovation / Spanish State Research Agency / European Regional Development Fund / European Union (Grants PGC2018-101370-B-I00, PID2021-128751NB-I00) and support from Science and Technology Facilities Council (CCP4-ARCIMBOLDO_LOW).

## Author contributions

Conceptualization: C.B., T.S., C.G., T.P.K. Methodology: C.B., T.S., C.G., N.G., U.B.S., I.U., T.P.K. Investigation: T.S., C.B., C.G., N.G., I.U., T.P.K. Visualization: T.S., C.B., C.G. Funding acquisition: N.G., I.U., T.P.K. Project administration: T.P.K. Supervision: U.B.S., I.U., T.P.K. Writing – original draft: C.B., T.S., C.G., N.G., T.P.K. Writing – review & editing: C.B., T.S., C.G., N.G., U.B.S., I.U., T.P.K.

## Competing interests

The authors declare no competing interests.

## Ethics

Authors mentioned contributed to the manuscript and roles and responsibilities were agreed. This research does not involve humans or animals.
