## [Peer Review File · Nature Communications]

SymProFold: Structural prediction of symmetrical biological assembliesREVIEWER COMMENTS

Reviewer #1 (Remarks to the Author):

A method for the assembly of S-layers using AlphaFold is presented. S-layers are 2D protein arrays that are formed by self-assembly of S-layer proteins. Currently, the method supports the assembly of S-layers that consist of single proteins. The method applies AlphaFold on the whole chain or domains to find two symmetry axes set to be parallel. ChimeraX commands and functions are used for assembly through domain and axes superposition. Overall, this is a nice application of AlphaFold for predicting structures of self-assemblies.

There is a link to a repository with code and models generated for a dataset presented in the paper. However, I didn't find any instructions on how to run SymProFold for a new sequence. There is a separate script for each benchmark case, is this customization needed?

How are steric clashes defined? is it the same irrespective of plddt? It seems that clashes in low plddt regions can be ignored

“Models in which a part of the protein chain passes incorrectly through the neighboring molecule are excluded “ - how? manually?

What is the MSA depth for the cases in the benchmark? is the success in the assembly related to having more sequences?

In the presented benchmark all the S-layers could be assembled. Are there cases where the method fails? What happens in the case of long linkers that connect protein domains?

Reviewer #2 (Remarks to the Author):

Pavkov-Keller et al report on a AF2-based method to predict symmetrical protein assembly. This topic is very popular in the community with multiple recent publications on the subject, and I somehow miss their discussion and the comparison in the manuscript. Please see a non-exhaustive list below. Please clearly present the novelty. I also miss a proper validation of the method on protein assemblies that were not a part of AF-multimer training set. It may simply happen that AF2 memorizes symmetrical arrangements of the training set. Please specifically consider validation of the method on non-crystallographic symmetries, including helical cases. Please consider cyclic symmetries of order higher than 6. Please discuss ambiguity of the symmetry group formation and compare results with solution experiments and possibly MD.

major points:

1)

You only cite one symmetry-prediction paper based on AF2 (21), however, there are multiple others. Please carefully review them, cite, and compare to the proposed method:

-- Li, Ziyao, et al. "Uni-Fold Symmetry: harnessing symmetry in folding large protein complexes." bioRxiv (2022): 2022-08.

-- Shor, Ben, and Dina Schneidman-Duhovny. "CombFold: predicting structures of large protein assemblies using a combinatorial assembly algorithm and AlphaFold2." Nature Methods (2024): 1-11.

-- Bryant, Patrick, et al. "Predicting the structure of large protein complexes using AlphaFold and Monte Carlo tree search." Nature communications 13.1 (2022): 6028.

APA

-- Gao, Mu, et al. "AF2Complex predicts direct physical interactions in multimeric proteins with deep learning." Nature communications 13.1 (2022): 1744.

-- Schweke, Hugo, et al. "An atlas of protein homo-oligomerization across domains of life." bioRxiv (2023): 2023-06. (here they used Ananas to complete cyclic symmetries)

2)

The symmetry identification is a well-recognized problem. Please compare your method (angle identification) and the scoring procedure to already published approaches above, and also to the symmetry identification method Ananas,

<https://doi.org/10.1016/j.jsb.2018.04.004>

<https://doi.org/10.1016/j.jsb.2018.05.005>

Why the angle-based scoring is better or worse than RMSD? Could you please discuss different strategies?

3) Please only validate your method on protein structures that were not used in training AlphaFold multimer (you can use PDB deposition data as a check). Please additionally provide test cases with a significant homology difference from the PDB training set used in AF2 (ideally less than 30%). It may simply happen that AF2 memoized assembly geometry from the training set.

Minor:

-- please correctly provide reference 21

Reviewer #3 (Remarks to the Author):

Predicting higher symmetry assemblies often requires substantial computational resources or is not possible using machine-learning-based methods like AlphaFold. Moreover, experimental structural characterization of these assemblies, like S-layers, poses a challenge, as traditional techniques such as X-ray crystallography and electron microscopy may not suffice due to the size of the complexes.

In their study, Buhlheller & Sagmeister et al. introduce SymProFold, an automated pipeline designed for predicting higher symmetrical assemblies such as S-layers and viral capsids. SymProFold only requires sequential information of the proteins as input, identifying sub-chains using the 'Domain_Separator' tool that was also developed by the authors. These sub-chains are then assembled into multimers, involving both sub-chains and full-length proteins. Subsequently, the symmetric complexes, ranging from dimers to hexamers, undergo quality assessment based on pTM, ipTM, and clashes, alongside the investigation of their symmetry (fold).

SymProFold then clusters the complexes according to their binding interfaces by constructing interface matrices (distograms) with a 10 Å Ca distance cutoff. Correlation coefficients of the distograms for the complexes to be investigated are calculated and the Louvain method is applied to identify interface (axis) clusters. Via superposition of the resulting complexes, SymProFold generates fully assembled models of the higher symmetry complexes (S-layers and virus capsids).

The efficacy of SymProFold was evaluated using a dataset consisting of 19 annotated S-layer proteins (SLPs), three non-SLPs, and a viral capsid from the Odonata-associated circular virus 21. The accuracy of the S-layer predictions was compared to available experimental data, demonstrating a high correlation with a 5% average difference in cell parameters. Moreover, SymProFold predicted the viral capsid symmetry and the symmetry of oligomers aligning with known literature findings for these complexes.

SymProFold is a very interesting method and will most likely be of interest, not only to those working on large supramolecular assemblies, but also those working on symmetric oligomers, especially since AlphaFold cannot identify the k-fold symmetry natively, and also has limits on the sizes of complexes it can model. Generally the manuscript is well-written, but it should undergo proofreading to address spelling and grammar errors,

I have two main concerns. First is that as described by the authors, the pipeline seems to be too stringent (will fail to assemble anything if certain conditions are not met). This may hinder the wide use of the pipeline, especially if it requires much manual intervention.

Secondly, while the authors have stated that there is a paucity in experimental data on S-layers, I'm wondering whether there are experimental data that would at least help verify some of the predicted interaction surfaces (or whether the authors could collect some of these themselves). This would make the paper much stronger, in my opinion.

Other aspects should be improved, as described below, and therefore I would suggest major revision:

1. Are there other experimental data that would help validate the predicted models aside from experimental unit cell size? Maybe some of the interfaces have already been described in the PDB?

2. P. 5: 'The symmetry complexes are then filtered based on a series of quality criteria, including the number of clashes.'

The filtering step should explicitly mention the specific criteria, rather than referring to a 'series of criteria.'

3. The authors often mention the score 'pTM+ipTM' when assessing complex quality (e.g., in Fig. 2). However, it is important to clarify that the actual score used is the model confidence, which is calculated as $0.8 \cdot ipTM + 0.2 \cdot pTM$. While this calculation method is explained in the 'Methods' section, it is not clear in the 'Results' or 'Discussion' sections. Also, could the authors also highlight the differences in ipTM+pTM values between models derived from full-length proteins and subchains? The supplementary data indicates instances where subchains successfully identified symmetry that full-length proteins did not. Could the authors speculate on potential reasons for this observation?

4. P. 6.: '...different steps of SymProFold were used to manually build the models of these S-layers...' It is not clear which specific procedures these "different steps" include.

5. Figure 4: Are these best scoring assemblies? Or was only one supramolecular assembly possible in all the cases? If so, please indicate that this is the case.

6. P. 10.: This part lacks the reference to the figure in the supplementary.

7. Paragraph 'Prediction of viral capsids': Incorrect figure referencing (should be Fig. 5)

8. The paragraph titled 'Prediction of viral capsids' describes the prediction of the viral capsid of Odonata-associated circular virus 21. However, it is lacking the assessment of the quality of this prediction, particularly considering that the maximum model confidence found is 0.72. If possible, the prediction should also be compared to any available experimental data.

9. In the section "Prediction of symmetrical Oligomers with only one Symmetry Axis", my takeaway was that SymProFold can predict the k-fold symmetry of these oligomers, but cannot assemble the models? Is this correct? If not, perhaps some more explanation is needed.

10. P. 11:; 'SymProFold identified a 3-fold symmetry complex with good scores (Figure S29).' The definition of a 'good' score is not clear. It would be more informative to mention absolute values.

11. Page 12:- "Due to the considerable size diversity of SLPs, some outreach 1000 amino acids" : replace "outreach" with exceed

12. Page 13: 'SymProFold predictions enable the development of computational models to simulate S-layers' behavior under different conditions.'

It is unclear how this is achievable, as the used modeling approaches do not consider 'conditions'. The statement requires further explanation.

13. It would be interesting to know why the test set (19 annotated S-layer proteins (SLPs), three non-SLPs, and a viral capsid from the Odonata-associated circular virus 21) was chosen and how it was selected.

14. Page 16;: 'whether the associated rotational symmetry axis is uniform within a tolerance range'. The specific tolerance range used in this step is not mentioned.

15. If the prediction cannot be performed using standard parameters of SymProFold, the authors refer to a 'manual parameter setting' of the pipeline (e.g. page 19; line 12). However, the process of performing this manual parameter setting is not clearly described.

16. There appears to be a discrepancy in the cutoff values used for filtering predictions. For instance, one section mentions -"Only homodimer predictions with a minimal score of 0.3 ipTM+pTM are further processed", while another section states that ""Models with an ipTM+pTM score of ≥ 0.20 and existing interfaces between the monomers are treated as symmetry complexes."

17. The authors mention- "With respect to the complete protein chain, relaxed models with at least 3.0 clashes per 100aa (and 60.0 per 100aa for unrelaxed) are excluded. Additionally, in all possible sequence sections of length 200aa, only 6.0 clashes per 100aa for relaxed models (and 120.0 per 100aa for unrelaxed) are allowed". Typically, the final structure should not exhibit any clashes involving main chain atoms.

18. The 'Computing requirements' section lacks details about the settings for the prediction of the complexes, including parameters like the number of recycles.

19. The authors should add documentation to the cloud repository. Without this documentation, it was not feasible to test SymProFold.

We would like to thank all reviewers for their careful reading and revision of our manuscript. Their valuable input has significantly improved the quality of the manuscript. We have addressed all points thoroughly. In addition, we provide two new crystal structures that support the SymProFold predicted S-layer interfaces of *Viridibacillus arvi* (PDB 9FS9) and *Methanococcus voltae* (PDB 9FSA).

We are submitting the revised main manuscript, along with the newly organized supporting information, which we subdivided into Supplementary Sections 1-11 for better clarity. For the newly presented crystal structures, we have provided coordination files (PDB), reflection files (MTZ), and validation reports. A detailed tutorial including source codes and calculated models is available at the GitHub repositories (<https://github.com/symprofold>).

Reviewer #1

"A method for the assembly of S-layers using AlphaFold is presented. S-layers are 2D protein arrays that are formed by self-assembly of S-layer proteins. Currently, the method supports the assembly of S-layers that consist of single proteins. The method applies AlphaFold on the whole chain or domains to find two symmetry axes set to be parallel. ChimeraX commands and functions are used for assembly through domain and axes superposition. Overall, this is a nice application of AlphaFold for predicting structures of self-assemblies."

We appreciate the feedback from reviewer #1 and addressed each concern thoroughly.

#1.1

"There is a link to a repository with code and models generated for a dataset presented in the paper. However, I didn't find any instructions on how to run SymProFold for a new sequence."

We added a detailed installation guide and an extensive tutorial, which can be found on our GitHub repository (with sample data or the possibility of calculating an S-layer assembly from a sequence; the user is required to have an AlphaFold-Multimer installation):

<https://github.com/symprofold>

The specific steps of the SymProFold pipeline are explained using the example of the S-Layer gene A0A1M5ZCF8 from *Vibrio aerogenes*. Additionally, we have provided an installation guide and tutorial on our GitHub repository for the 'Domain_Separator' tool for identifying structural domains in monomeric proteins or homo-multimer complexes.

We updated the Methods part "Data and code availability" accordingly.

#1.2

"There is a separate script for each benchmark case, is this customization needed?"

A separate script for each benchmark case is available for reproduction purposes of our results, thereby allowing the calculation and reproduction of the exact same assembly as the published S-layers.

In detail:

All combinations of symmetry complexes are tested for assembly formation and scored based on quality. For each combination, only one model (a reduced assembly model) is calculated and stored. Calculating all types of assembly models, including the primitive unit cell with symmetry information and the 3x3 assembly (both full-length and variant without termini), would be impractical due to the computational power and storage capacity required (~50 GB of data) for all combinations.

The final assembly script calculates all types of assembly models for the best combinations only. It takes a set of parameters (symmetric complexes, alignment domain) that results in a good quality score as input.

#1.3

"How are steric clashes defined? is it the same irrespective of plddt? It seems that clashes in low plddt regions can be ignored "

Steric clashes are determined by the ChimeraX function command *clashes* using standard parameters. The cutoff criteria are described in the Methods section "Predictions of Oligomer Sets and Filtering."

For further clarification, a new paragraph has been added to the supplementary information Supplementary section 3 "Prediction of Oligomer Sets and Filtering" providing detailed explanations for the calculation of overlaps and the exclusion of unusually high fractions of intermolecular beta sheets. Additionally, we included a description of the automated determination of N- and C-termini, which are excluded from the clash score determination (see Supplementary section 3: "Prediction of Oligomer Sets and Filtering"). Other regions with high flexibility (e.g., loops) are not excluded in order to provide complete models.

Weighting the clashes according to their prediction confidence (pLDDT) is not performed. The final assemblies are ranked (not filtered) by the combined quality score ($\text{score}_{\text{quality}} = \text{score}_{\text{clash}} + \text{score}_{\text{bend}}$). For assessing the superposition of the assembled S-layer with the combined quality score, confidence weighting is challenging since pure confidence weighting leads to the effect that poorer predictions (with many low pLDDT ranges) tend to be favored over better predictions. It should also be noted that prediction errors in low pLDDT linker regions can lead to incorrect relative positioning of domains, resulting in severe clashes in high pLDDT regions. As the final $\text{score}_{\text{quality}}$ is used for ranking rather than filtering, a high value highlights issues in the supramolecular assembly that need to be addressed.

“Models in which a part of the protein chain passes incorrectly through the neighboring molecule are excluded “ - how? manually?”

Protein chains passing incorrectly through neighboring molecules and forming false intermolecular β -sheets represent a recurring issue in AlphaFold-Multimer predictions (0.1%-1% of the calculations conducted in this study). To address this issue, SymProFold contains a built-in algorithm that automatically excludes predictions with unusually high fractions of intermolecular β -sheets. This is now described in detail in the Supplementary Section 3 “Prediction of Oligomer Sets and Filtering”.

#1.4

“What is the MSA depth for the cases in the benchmark?”

All calculations were carried out using an AlphaFold-Multimer 2.3 installation in its standard configuration and with full MSA depth; we did not use an external MSA server. The exact installation and workstation specifications are described in the updated Methods section “Computing Requirements”.

"is the success in the assembly related to having more sequences?"

We do not have such a comparison, as all our calculations were carried out using the complete AlphaFold database. Since S-layer proteins tend to have relatively low sequence homology, we recommend using the complete and most recent AlphaFold database.

#1.5

"In the presented benchmark all the S-layers could be assembled. Are there cases where the method fails?"

During the development and benchmarking of the algorithm, we encounter cases for which the method fails, as described in the Discussion paragraphs 2 and 3.

In detail, sequences exceeding 1000aa are problematic when using AlphaFold Multimer 2.3 since they exceed the available computing power resulting in non-reliable predictions. Additionally, putative S-layer sequences for which only one rotational axis could be found, due to for instance very weak interactions, did not produce fully assembled models. Also, incorrect or incomplete annotations seem to occur regularly for S-layer protein sequences. For sequences for which only one symmetry axis could be found, we propose the following explanations:

a) Incorrect annotation of the putative S-layer sequence:

The protein might not be an S-layer protein and cannot assemble into a 2D layer. This is comparable to the results section "Prediction of Symmetrical Oligomers with only one Symmetry Axis", where we tested SymProFold on proteins known to form defined symmetrical oligomers but which are not S-layer proteins. In these cases, SymProFold stopped at a certain step and did not result in an assembled layer.

b) The S-layer is composed of more than one protein:

Heterolayers of two or more S-layer proteins can occur in nature but are often not well described or correctly annotated. If a sequence does not output a reliable SymProFold model, it could also be due to a second S-layer protein that builds the layer.

c) Post-translational modifications or cleavage:

Some S-layers are post-translationally modified or cleaved (e.g. SlpA from *Clostridium difficile*, Lanzoni-Mangutchi et al. 2022). Currently, we cannot extract this information or account for the impact of such modifications, which may lead to incorrect predictions.

#1.6

"What happens in the case of long linkers that connect protein domains?"

The algorithm treats all linkers equally. However, in special cases with extremely long domain linkers, careful revision of the results is advised. In our current dataset, we did not encounter any example where such revision would have been necessary.

Reviewer #2

“Pavkov-Keller et al report on a AF2-based method to predict symmetrical protein assembly. This topic is very popular in the community with multiple recent publications on the subject, and I somehow miss their discussion and the comparison in the manuscript. Please see a non-exhaustive list below.”

We thank reviewer #2 for the valuable annotations and we adjusted the manuscript accordingly.

#2.0.1

Please clearly present the novelty.

In agreement with reviewer #2 we expanded the introduction by highlighting novel features of SymProFold in comparison to currently available methods. More detailed explanation please see below at **#2.1**.

#2.0.2

I also miss a proper validation of the method on protein assemblies that were not a part of AF-multimer training set. It may simply happen that AF2 memorizes symmetrical arrangements of the training set. Please specifically consider validation of the method on non-crystallographic symmetries, including helical cases.

We agree with reviewer #2 and added a short explanation in the introduction section, highlighting that none of the shown structures (or fragments of the structures) are part of the AlphaFold-Multimer training set:

“The assemblies of 19 new S-layers from bacteria and archaea as well as a viral capsid are predicted, which were not part of the AlphaFold-Multimer training set (Supplementary Table 2).”

In detail, the following structures (and fragments of the structures) are not included in the training set of the AlphaFold-Multimer software:

Desulfurococcus mucosus (E8R795); Desulfurococcus amylolyticus (I3XTG6); Sporosarcina ureae (Q0VJW4); Aeromonas salmonicida (P35823); Vibrio aerogenes (A0A1M5ZCF8);

Vibrio quintilis (A0A1M7YYM3); *Aneurinibacillus thermoaerophilus* (Q6TL21); *Viridibacillus arvi* (slp1, A0A0K2Z0V7); *Paenibacillus naphthalenovorans* (A0A0U2M877); *Methanococcus vannielii* (A6URZ5); *Methanococcus voltae* (Q50833); *Thermococcus camini* (A0A7G2D8K1); *Thermococcus thio还原ens* (A0A0Q2M111); *Pyrococcus abyssi* (Q9V0N3); *Thermoanaerobacter kivui* (P22258); *Brevibacillus brevis* (P06546); *Phocaeicola vulgatus* (A0A5P3AY64) and *Corynebacterium glutamicum* (Q2VRQ3).

Regarding the S-layer of *Bacillus anthracis* (EA1, P94217), the PDB entry (PDB 8OPR) was released on November 8, 2023, which was after the AlphaFold-Multimer training cutoff on September 30, 2021. Technical note of AlphaFold v2.3.0: https://github.com/google-deepmind/alphafold/blob/main/docs/technical_note_v2.3.0.md

Please also consider our response to **#2.3**. In the new Supplementary Table 2 we additionally show with an HHpred analysis, that in the presented test cases on average, only for ~ 17% of the sequence a homologues relationship was found. Often, this is the part responsible for the cell-wall binding, with the exception of *M. voltae* and *P. abyssi*, both showing similarity to the dimeric structure of 3U2G of *M. acetivorans*. This crystal structure shows an overall similar domain arrangement at the 2-fold axis in the asymmetric unit (non-crystallographic dimer), but also has some crucial differences, that we highlight in the Supplementary Figure 27 and is also described in the manuscript in the section “Exploring Novel Features Revealed by Predicted S-Layer Assemblies”. In addition, we present a newly obtained crystal structure of this anchor at the 2-fold axis interface, not present in the crystal structure of 3U2G, which is important for the domain arrangement at this interface. We added a new paragraph in the section “Validation of Models with Experimental Data” with the new data.

Regarding the raised concern of helical cases, we would like to refer to **#2.0.3**. We specifically designed SymProFold to predict 2-dimensional supramolecular assemblies as found in S-layers and to some extent in virus capsids. Therefore, we did not consider helical cases for validation.

#2.0.3

Please consider cyclic symmetries of order higher than 6.

In general, with only minor adjustments, the SymProFold algorithm could also handle symmetries higher than six. Nevertheless, to date, there are no experimental data available for repetitive 2D assemblies exhibiting symmetries above six. Since we prepared SymProFold

specifically for the calculation of S-layers and viral 2D assemblies, we believe that extending the method to higher symmetries is only relevant if supported by experimental evidence.

Additionally, theoretical higher-order symmetries are harder to distinguish. For example, the angle between symmetry mates in an 8-fold symmetry axis is 45° , whereas in a 9-fold symmetry axis, it is 40° , resulting in an angular difference of only 5° .

#2.0.4

Please discuss ambiguity of the symmetry group formation and compare results with solution experiments and possibly MD.

We appreciate the comment of reviewer #2 and provide additional high-resolution structural data thereby successfully validating SymProFold derived models. The newly presented crystal structures, together with the published low-resolution experimental data (Table 1), emphasize the reliability of SymProFold determined symmetry groups.

Due to the insolubility of full-length S-layers, we used soluble domains of the S-layer assemblies for high-resolution structure determination. These domains were chosen according to SymProFold derived models. These two new structures, 9FSA and 9FS9, are incorporated in the updated Figure 4 in the section “Validation of Models with Experimental Data”, confirming our proposed SymProFold models.

We value the comment of Reviewer#2 concerning the Molecular Dynamic experiments. MD simulations of the fully assembled layer could give further insight into the stability of the arrangement and interfaces. Due to the tremendous size of an assembled layer (minimum of nine unit cells, exceeding up to 500.000+ atoms with solvent), MD simulations require extensive computational power. In our very short predictions (< 10 ns), we were not able to observe any relevant results.

In detail:

For a specific sequence SymProFold outputs a plot that includes all computationally possible symmetry groups to form an assembly. Favored symmetry groups can be detected via their high ipTM+pTM values. Using the currently available published low-resolution data of S-layers enabled a successful confirmation of SymProFold determined symmetries (Table 1). In addition to the published data, we furthermore provide two new crystal structures of domains

that are forming the symmetry axis of *Viridibacillus arvi* and *Methanococcus voltae* S-layer assemblies. The low RMSD values of the experimental structures to the models (0.65 Å and 1.38 Å, respectively), support that SymProFold offers a precise and solid calculation of S-layer symmetries.

#2.1

You only cite one symmetry-prediction paper based on AF2, however, there are multiple others. Please carefully review them, cite, and compare to the proposed method:

We agree with reviewer #2 and have added an additional paragraph to the introduction discussing the currently available methods for calculating models of oligomeric protein complexes and highlighting the differences to SymProFold.

The SymProFold pipeline is specialized on calculating supramolecular assemblies, like two-dimensional S-layer assemblies or virus capsid arrangements exhibiting 2-, 3-, 4-, or 6-fold rotational symmetry axes, repeating pattern (unit cell) and is consisting of one protein. To our knowledge, SymProFold is currently the only method for determining supramolecular assemblies and virus capsids using sequential information only, without applying prior knowledge about symmetry or oligomerization state.

“AlphaFold has been extensively used to predict individual virus proteins^{21,22} and a method to combine AlphaFold monomer predictions with symmetric all-atom docking simulations to predict cubic complexes is available²³. Several methods are currently available for the prediction of large oligomeric complexes which require besides the sequence also additional information as input to ensure a reliable output. Depending on the method used in addition to the protein sequence, either stoichiometry information^{24–26} or symmetry group information²⁷ are needed. Schweke et al. present a pipeline for the calculation of cyclic homo-oligomers using sequential information and the program AnAnaS²⁸ for the identification of symmetries, concentrating on cyclic symmetries and dihedral or cubic groups.”

#2.2

The symmetry identification is a well-recognized problem. Please compare your method (angle identification) and the scoring procedure to already published approaches above, and

also to the symmetry identification method Ananas,

<https://doi.org/10.1016/j.jsb.2018.04.004>

<https://doi.org/10.1016/j.jsb.2018.05.005>

Why the angle-based scoring is better or worse than RMSD? Could you please discuss different strategies?

SymProFold compares the ipTM+pTM score for different angles belonging to different symmetry orders. High scores indicate a certain order of symmetry.

Symmetry identification with AnAnaS [1,2] was done by Schweke et al. 2024 on homodimer models. AnAnaS calculates an RMSD value for different symmetries. Ideally, the prevailing symmetry in a perfectly symmetrical model would lead to an RMSD of 0. The RMSD values calculated with AnAnaS for our predicted models correlate well with the ipTM+pTM values for full oligomers (e.g. a full tetramer consisting of 4 subchains). Low RMSD values correlate with high ipTM+pTM values and vice versa. Using the RMSD metric instead of ipTM+pTM for AlphaFold-Multimer predictions probably leads to comparable results for the S-Layer models published in this study. However, it can be observed that within an AlphaFold-Multimer run, the RMSD values of the output models appear to be more broadly distributed than the corresponding ipTM+pTM values. The distinction of different symmetries via the RMSD appears to be more difficult for predictions with AlphaFold3 (Abramson, J., et al. 2024) due to a higher intrinsic symmetry compared with AlphaFold-Multimer predictions.

[1] Pagès G, Kinzina E, Grudinin S. Analytical symmetry detection in protein assemblies. I. Cyclic symmetries. *J Struct Biol.* 2018 Aug;203(2):142-148. doi: 10.1016/j.jsb.2018.04.004.

[2] Pagès G, Grudinin S. Analytical symmetry detection in protein assemblies. II. Dihedral and cubic symmetries. *J Struct Biol.* 2018 Sep;203(3):185-194. doi: 10.1016/j.jsb.2018.05.005.

#2.3

"3) Please only validate your method on protein structures that were not used in training AlphaFold multimer (you can use PDB deposition data as a check). Please additionally provide test cases with a significant homology difference from the PDB training set used in

AF2 (ideally less than 30%). It may simply happen that AF2 memoized assembly geometry from the training set."

We appreciate the remark of reviewer #2 and have added an additional Supplementary Table 2 analyzing the sequential and structural homologies of the presented test cases to the PDB database using RCSB sequence similarity search and HHpred, which includes a structural prediction.

In summary, the RCSB sequence similarity search reveals that 14 of the 18 test cases do not have matching results in the PDB. Two out of 18 test cases (*T. kivui* (P22258) and *B. brevis* (P06546)) have sequence similarities below 30% to their best hits. *S. ureae* (Q0VJW4) reveals a sequence identity of 37% to 8BYS but includes only residues 9-78, forming the cell wall binding domain, which is not relevant for the formation of the assembly. *P. naphthalenovorans* (A0A0U2M877) has a sequence similarity of 34% to 6CWC, including residues 1-172, due to the presence of an SLH cell wall binding domain. Additionally, we performed an analysis using HHpred (<https://toolkit.tuebingen.mpg.de/tools/hhpred>) showing that only short regions of the S-layer proteins (9-26%) could be aligned, often solely including the cell wall binding domain.

Supplementary Table 2. Homology analysis of the presented 18 test cases by RCSB sequence similarity search and HHpred. RCSB sequence similarity search outputs matching results for only four out of the 18 test cases. HHpred analysis shows that only 9-26% of the input sequences could be aligned.

Organism	Uni Prot ID	RCSB sequence similarity search			HHpred			Input sequence
		Best Hit UniProt ID	SS %	Aligned region	Best Hit	E-value	Aligned region	
C. glutamicum	Q2VRQ3	No result			7WOO_I	80	11%	1-491 (FL)
A. salmonicida	P35823	No result			6P1E_B	3	16%	22-502 (*)
V. quintilis	A0A1M7Y YM3	No result			6P1E_B	2.1	20%	22-642 (*)
V. aerogenes	A0A1M5Z CF8	No result			6P1E_B	1.8	19%	22-646 (*)
T. camini	A0A7G2D 8K1	No result			3U2G_A	0.053	12%	1-489 (FL)
P. vulgatus	A0A5P3A Y64	No result			4ZXQ_D	0.011	13%	1-1100 (FL)
M. vannielii	A6URZ5	No result			3U2G_A	0.0054	19%	1-567 (FL)
T. thio还原ens	A0A0Q2M 111	No result			3U2G_A	0.00035	15%	1-607 (FL)
V. arvi	A0A0K2Z0 V7	No result			4UIC_A	0.000034	25%	1-1016 (FL)
T. kivui	P22258	3PYW	27%	5-129	8ACQ_A	0.000012	26%	1-762 (FL)
B. brevis	P06546	6CWN	29%	1-172	6CWM_A	1.2e-7	23%	1-1053 (FL)
D. mucosus	E8R795	No result			7PTR_E	9.9e-8	11%	1-904 (FL)

A. thermoaerophilus	Q6TL21	No result			5H3K_A	5.7e-8	9%	1-738 (FL)
D. amylolyticus	I3XTG6	No result			7PTR_E	4.7e-11	15%	1-898 (FL)
M. voltae	Q50833	No result			3U2G_A	4.5e-13	19%	1-576 (FL)
S. ureae	Q0VJW4	8BYS	37%	9-78	4UIC_A	1.1e-15	15%	1-1097 (FL)
P. abyssi	Q9V0N3	No result			3U2G_A	2.7e-17	18%	1-604 (FL)
P. naphthalenovorans	A0A0U2M877	6CWC	34%	1-172	4UIC_A	8.5e-37	15%	1-1053 (FL)

(*) without signal sequence

#2.4

Please correctly provide reference 21

The preprint reference (as of submission Nov. 2023) was updated with the published version as of Dec. 2023.

Reviewer #3

“...SymProFold is a very interesting method and will most likely be of interest, not only to those working on large supramolecular assemblies, but also those working on symmetric oligomers, especially since AlphaFold cannot identify the k-fold symmetry natively, and also has limits on the sizes of complexes it can model. Generally the manuscript is well-written, but it should undergo proofreading to address spelling and grammar errors...”

We gratefully acknowledge the detailed statement of reviewer #3 and we addressed all the addressed points.

#3.0.1

I have two main concerns. First is that as described by the authors, the pipeline seems to be too stringent (will fail to assemble anything if certain conditions are not met). This may hinder the wide use of the pipeline, especially if it requires much manual intervention.

Upon remark of reviewer #3, we added a short description of the conditions that induce a termination of the pipeline, which can also be adapted manually if needed. Generally, the pipeline tests all reasonably possible assemblies and ranks them by the overall score. In non-optimal cases, the pipeline does not terminate prematurely, but the results will have a low overall score. We updated the discussion part with the following paragraph:

“The SymProFold pipeline terminates if at least one of the following cases is true: the axis tilt is above 45° (SymProFold supports 2-, 3-, 4- or 6-fold rotational symmetry), the spatial gap between the predicted symmetry complexes is too large (and they cannot be connected), the deviation between the calculated lattice constant (unit cell) and the model (by superposition) is too large, or a symmetry axis complex does not have a rigid folding unit.”

A symmetry complex is considered as "having no rigid folding unit" if in a chain all residues have an error estimate (predicted RMSD) > 5Å.

#3.0.2

Secondly, while the authors have stated that there is a paucity in experimental data on S-layers, I'm wondering whether there are experimental data that would at least help verify

some of the predicted interaction surfaces (or whether the authors could collect some of these themselves). This would make the paper much stronger, in my opinion.

Please see the answer to comment #3.1

#3.1

1. Are there other experimental data that would help validate the predicted models aside from experimental unit cell size? Maybe some of the interfaces have already been described in the PDB?

We agree with the importance of experimentally derived structures as suggested by reviewer #3 and provide two additional crystal structures of S-layer domains, that were collected after the initial submission: the 4-fold symmetry axis of *Viridibacillus arvi* (PDB: 9FS9) and the dimer anchor of *Methanococcus voltae* (PDB: 9FSA). Both deposited structures are currently on HOLD and not accessible on the PDB RCSB databank, but we uploaded the structure and reflection files in addition to the reviewed manuscript. Both experimentally derived structures are in good alignment with the presented models thus confirming SymProFold predicted interfaces (*V. arvi* RMSD: 0.65 Å; *M. voltae* RMSD: 1.38 Å; see results section: “Validation of Models with Experimental Data”). In addition, we updated Figure 4 with the new data.

Additionally, we analyze SymProFold derived structures with published experimental data in the results section “Validation of Models with Experimental Data”. This section offers validation and verification of SymProFold models by available low-resolution electron microscopy and AFM data, as well as high-resolution crystal structures. Supplementary section 8 “Interface comparison of experimental structures” furthermore provides additional detailed comparisons of currently available experimental structures with SymProFold derived models.

In detail:

In addition to our in-house experiments, the available high-resolution data is very scarce. The crystal structure of C-terminal DUF1608 domain of the S-layer protein of the archaea *M. acetivorans* (PDB code: 3U2H) reveals a homolog binding interface to the SymProFold model of the S-layer of archaea *M. vannielii* (see Supplementary Figure 27).

Recently, two new high-resolution structures of S-layer assemblies were published. First, the assembled structure of EA1 from *Bacillus anthracis* was described by Sogues et al. in 2023.

This high-resolution model agrees well with our proposed model and shows the same two main binding interfaces, aligning well with experimentally derived high-resolution data (see Supplementary section 8, Supplementary Figure 25).

Second, in April 2024, Kuegelgen and Bharat deposited several cryo-EM-derived high-resolution structures of the S-layer protein from *Nitrosopumilus maritimus* to the PDB databank (PDB: 8C8M, Kuegelgen et al. 2024), exhibiting a fully assembled 6-fold axis. The SymProFold derived assembly model of this 1734 amino acid long S-layer protein reveals similar interfaces although this structure was not part of the AlphaFold training set. More detailed structural comparisons are shown in Supplementary Figures 25 and 26 of Supplementary Section 8 and described in the results section “Validation of Models with Experimental Data”.

#3.2

P. 5: ‘The symmetry complexes are then filtered based on a series of quality criteria, including the number of clashes.’

The filtering step should explicitly mention the specific criteria, rather than referring to a ‘series of criteria.’

In agreement with reviewer #3 we provide a more detailed explanation of the criteria used for filtering the symmetry complexes as shown in the adjusted paragraph below. Please also refer to the paragraph “Prediction of Oligomer Sets and Filtering” in the Method section.

*“The symmetry complexes are then filtered based on i) rotational symmetry, ii) a weighted model confidence score ($0.8*ipTM+0.2*pTM$, further referred to as $ipTM+pTM$ score) of at least 0.2, iii) the number of clashes, and iv) no unusually high fraction of intermolecular β -strands (Supplementary section 3-5).”*

#3.3

*3. The authors often mention the score ‘ $pTM+ipTM$ ’ when assessing complex quality (e.g., in Fig. 2). However, it is important to clarify that the actual score used is the model confidence, which is calculated as $0.8*ipTM+0.2*pTM$. While this calculation method is explained in the ‘Methods’ section, it is not clear in the ‘Results’ or ‘Discussion’ sections.*

We agree with reviewer#3 and added a clarification at the first occurrence in the Results section, that the weighted model confidence score is calculated as $0.8 \cdot ipTM + 0.2 \cdot pTM$ and referred to as $ipTM + pTM$ score.

Also, could the authors also highlight the differences in $ipTM+pTM$ values between models derived from full-length proteins and subchains?

We appreciate the comment of reviewer#3 and added an additional sentence referring to Figure 2, concisely noting the differences in $ipTM+pTM$ scores between full-length (FL) and other subchain predictions:

“Compared to full-length predictions, other subchain predictions can result in higher $ipTM+pTM$ scores due to the omission of domains connected via flexible linkers and their introduced uncertainty reflected in lower scores.”

The following paragraph, with an additional figure clarifying these differences, was added to the Supplementary Section 5:

*“The clustered symmetry complexes with the highest $ipTM+pTM$ score can, but do not need to be FL predictions. The difference in the model confidence score originates mainly from the difference in the $ipTM$ score due to its predominant proportion of 80%. The $ipTM$ score is an accuracy estimate calculated from the errors in position between inter-chain residues. A prediction resulting in a relatively rigid assembly and clearly defined domain arrangement is equivalent to small inter-chain residue errors, leading to a high $ipTM$ score. In contrast, domains with a flexible position relative to the assembly are characterized by larger inter-chain residue errors, leading to a low $ipTM$ score. Exemplified by the predictions of *A. salmonicida* the comparison is shown in Figure 2. Predictions with smaller subchains only containing the domains relevant for symmetric assembly give higher $ipTM+pTM$ scores than the FL predictions.”*

For example, the assembly around the first 4-fold symmetry axis of the *A. salmonicida* S-layer (Figure 2) is formed by binding sites of domain 1 and domain 2, resulting in a fixed domain arrangement of subchain 1-2. The full-length sequence prediction has a flexible linker between domain 2 and domain 3, enabling domain 3 to adopt several different positions, which is consistent with the lower $ipTM$ score. In addition, Figure 2 was updated with more graphical information about the Subchains and multimer predicitions.

The assembly around the second 4-fold symmetry axis is formed only by domain 3, resulting in a fixed domain arrangement and a high ipTM score. The position of domain 2 in the assembly of subchain 2-3 has more freedom of movement due to the flexible linker between subchain 2 and domain 3. This is consistent with the lower ipTM score of this prediction.

The supplementary data indicates instances where subchains successfully identified symmetry that full-length proteins did not. Could the authors speculate on potential reasons for this observation?

The necessity for subchain predictions can arise for different reasons:

Reason A: One symmetry complex is dominantly predicted

Often, full-length predictions result in one strongly favored symmetry complex that dominates over a second symmetry complex (i.e., only the “stronger” binding site is predicted). In many cases, the second symmetry complex appears when the domains involved in the stronger binding site are removed (resulting in a subchain without the dominant binding site).

Examples: *A. salmonicida* (P35823), *V. quintilis* (A0A1M7YYM3), *V. aerogenes* (A0A1M5ZCF8).

Reason B: System size exceeds A100 GPU

FL predictions with a high order of symmetry result in very large system sizes (greater than 3000-4000 amino acids). Several predictions with very large system sizes could not be calculated on the A100 GPU used in this study.

Examples of which the system size allowed full-length prediction of only 1 of 2 symmetry axes: *P. vulgatus* (A0A5P3AY64), *T. kivui* (P22258), *B. brevis* (P06546), *D. mucosus* (E8R795), *D. amylolyticus* (I3XTG6), *S. ureae* (Q0VJW4).

Examples of which the system size did not allow full-length prediction of a symmetry axis: *P. naphthalenovorans* (A0A0U2M877).

Reason C: Both symmetry axes are present in one full-length assembly, but the assembly is twisted

For some cases a full-length prediction results in an assembly containing both symmetry axes, which leads to an incorrectly twisted assembly in order to adapt both symmetry axes. In these cases, this model is filtered out by the filtering step ("Scoring of Symmetry Axis Complexes" in Figure 1) due to the twisting. Furthermore, using AlphaFold3 (Abramson, J., et al. 2024) we observe that twisted assemblies appear more often, as tested in May 2024.

Examples: *T. camini* (A0A7G2D8K1), *M. vanniellii* (A6URZ5), *V. arvi* (A0A0K2Z0V7), *M. voltae* (Q50833).

In this study, we only encountered one example (*C. glutamicum* (Q2VRQ3)) where both symmetry complexes were identified by full-length predictions, and the need for subchains would not have been necessary.

These examples with an explanation have been added to a new Supplementary Table 4 added to the Supplementary with an explanatory paragraph:

"Supplementary Table 4: Symmetry axes identified by full-length predictions. Full-length predictions can lead to an assembly model, but generally, subchain predictions are necessary. Reasons include A) one strongly favored symmetry center, B) too large system sizes, and C) twisted assemblies to produce both symmetry centers in one prediction. "

An additional sentence was also introduced in the main paper (Results, paragraph 1):

"The need for using subchains arises from three main reasons. Large systems (greater than 3000-4000 amino acids) exceed the computing power of the used hardware, one symmetry center is strongly favored and hinders reliable prediction of the second symmetry center, or an assembly is impossible due to twisted arrangements of full-length models (Supplementary Table 4)."

#3.4

4. P. 6.: '...different steps of SymProFold were used to manually build the models of these S-layers... 'It is not clear which specific procedures these "different steps" include.

We appreciate the remark of reviewer#3. The mentioned sentence was changed for more clarification:

“For p1 S-layers of EA1 from Bacillus anthracis and the main SLP from Bacillus licheniformis, an augmented set of single domain subchains and heterodimer predictions with the FL protein were used as described in the methods section (Figures S23, S24).”

An additional paragraph was added to the Supplementary Section 7: Additional steps in p1 SLPs, describing in detail the workflow which differs to the presented standard procedure (see below).

“Additional steps in p1 SLPs

To predict low symmetry assemblies the standard workflow of SymProFold was augmented by additional subchains and heterodimer predictions. For each domain a subchain was created and those were exhaustively predicted as heterodimers with the FL protein. Supplementary Figure 23A shows the FL model of EA1, an SLP from B. anthracis, with the single domain predictions with highest ipTM+pTM scores. The ipTM+pTM scores of all predictions are shown in Supplementary Figure S24.

Same steps are shown in the workflow of Figure 1: “Prefiltering”, “Subchain identification”, “Prediction of multimer sets”, “Scoring of symmetry axis complexes”, “Superposition and optimization”, “Parameter extraction”, “Assembly of Unit Cell”, but some of the steps are extended to cover also the situation of a p1 lattice. The prefiltering step remains unchanged. The standard set of subchains is extended by subchains covering 1 single domain each. In the prediction step, dimer predictions with FL and each single domain are performed. Example results for the dimer predictions of Bacillus anthracis (P94217) are shown in a scoring diagram depicted in Supplementary Figure 24. In the superposition step the layer is built by superposition and the quality score is calculated ($score_{quality} = score_{clash} + score_{bend}$), unit cell parameters are extracted.”

#3.5

5. Figure 4: Are these best scoring assemblies? Or was only one supramolecular assembly possible in all the cases? If so, please indicate that this is the case.

For all presented models and the unit cell dimensions described in Table 1, one representative of the ensemble of high-scoring models is shown. SymProFold tests all possible superpositions of symmetry complexes and calculates the quality score (described in “Methods – Parameter

Extraction”); $\text{score}_{\text{quality}} = \text{score}_{\text{clash}} + \text{score}_{\text{bend}}$) for each resulting assembly model. Among the results, there are typically several high-scoring models with similar $\text{score}_{\text{quality}}$ and lattice constants. We added a short explanation in the results section “Assessment of prediction quality”:

“The structures shown are representative for the calculated ensemble of SymProFold models exhibiting high quality scores (see “Methods – Parameter Extraction”).”

#3.6

6. P. 10.: *This part lacks the reference to the figure in the supplementary.*

We added a reference to Supplementary Figure 28 (Supplementary section 9 “Exploring Novel Features Revealed by Predicted S-Layer Assemblies”) in the results section “Exploring Novel Features of Predicted S-Layer Assemblies”.

#3.7

7. Paragraph ‘Prediction of viral capsids’: *Incorrect figure referencing (should be Fig. 5)*

We thank Reviewer #3 for the remark and we corrected the references accordingly.

#3.8

8. *The paragraph titled 'Prediction of viral capsids' describes the prediction of the viral capsid of Odonata-associated circular virus 21. However, it is lacking the assessment of the quality of this prediction, particularly considering that the maximum model confidence found is 0.72. If possible, the prediction should also be compared to any available experimental data.*

We generated a new plot (Supplementary Figure 22, Supplementary Section 6 “ipTM+pTM of predicted symmetry complexes”) showing the median ipTM+pTM score of our benchmark case and expanded the explanation in the "Assessment of Prediction Quality" chapter of the results section. We furthermore expanded the results section “Prediction of Viral Capsids“ by providing an analysis of the viral capsid model and available literature. Due to limited published data, we could only compare the capsid diameter of the presented capsid model of Odonata-associated circular virus and other members of the same *Smacoviridae* phylum (Krupovic et al. 2020).

In detail:

Regarding the viral capsid of Odonata-associated circular virus 21, the 3-fold symmetry axis is predicted with an ipTM+pTM score of 0.72 (Supplementary Figure 21) for the best-ranked model. The 2-fold symmetry axis is part of predictions with an ipTM+pTM score of 0.55 (Supplementary Figure 21) for the best-ranked model. Compared to other predictions of S-layers, the ipTM+pTM scores are within a typical range for predicted assembly models, albeit in the lower range. This information was also added to the "Prediction of Viral Capsids" paragraph.

“Compared to the median ipTM+pTM score (Supplementary Figure 22) of our benchmark cases (axis A: 0.80, axis B: 0.74), the virus example shows slightly lower ipTM+pTM values for axis A 0.72 and axis B 0.55, but are still within a typical range of the predicted models. “

#3.9

9. In the section "Prediction of symmetrical Oligomers with only one Symmetry Axis", my takeaway was that SymProFold can predict the k-fold symmetry of these oligomers, but cannot assemble the models? Is this correct? If not, perhaps some more explanation is needed.

This statement is correct: SymProFold did not generate assembly models for the presented examples since the mentioned proteins do not form supramolecular assemblies. For all three of the presented test cases, SymProFold stopped, and no 2D layer could be generated for proteins with only one rotational symmetry axis.

We chose these three examples with known rotational symmetry to test SymProFold on proteins that possess a rotational symmetry axis but should not be able to form a 2D layer. In all three cases, only one rotational symmetry axis was found, preventing 2D assembly formation. Consequently, the pipeline stopped as intended, and no layer was generated.

For YabJ (D4G3D4) and N9 neuraminidase (P03472), the SymProFold clustering algorithm identified only one rotational symmetry axis cluster. A second symmetry axis cluster, required to span a 2D assembly, was not available. For KaiC (Q8GGL1), the algorithm returned two clusters, but the rotational symmetry axes of both clusters coincided, making 2D layer formation impossible.

#3.10

10. P. 11;: *'SymProFold identified a 3-fold symmetry complex with good scores (Figure S29).'* The definition of a 'good' score is not clear. It would be more informative to mention absolute values.

We rephrased the sentence accordingly (see below). Summarizing, the 3-fold symmetry axis of YabJ was calculated with an ipTM+pTM score of 0.96 (Supplementary Figure 29) for the best-ranked model, as expected since this structure was part of the AlphaFold-Multimer training set. This score is, compared to predictions of benchmark S-layers (Supplementary Figure 22), higher than the median score for axis A. However, a second symmetry axis B that would be needed for a 2D assembly, was not predicted for YabJ.

"SymProFold identified a 3-fold symmetry complex with a high ipTM+pTM of 0.96, which is higher than the median score of the benchmark cases probably due to the structure's association in the AlphaFold-Multimer training set (Supplementary Figure 33)."

#3.11

11. Page 12:- *"Due to the considerable size diversity of SLPs, some outreach 1000 amino acids": replace "outreach" with exceed*

The sentence was corrected accordingly.

#3.12

12. Page 13: *'SymProFold predictions enable the development of computational models to simulate S-layers' behavior under different conditions.'* It is unclear how this is achievable, as the used modeling approaches do not consider 'conditions'. The statement requires further explanation.

We appreciate the comment of reviewer #3 and have rephrased the paragraph in the Discussion section as follows:

"To gain insights into these dynamic processes, the SymProFold predicted assemblies can serve as a starting point to identify the required interactions within the lattice, perform molecular dynamic studies and investigate binding events with ligands or receptors. S-layers often represent the outer most layer, therefore environmental conditions could impact their

structure and functionality. Using the predicted assemblies in combination with computational methods, the influence of e.g. pH changes on the S-layer could be investigated. “

Summarizing, the S-layer models can be used to determine biophysical properties such as the electrostatic potential, as illustrated in the updated Supplementary Figure 30, which shows the cell-facing side, the surface-exposed regions, and the pores of the calculated assemblies. Furthermore, SymProFold outputs detailed structural information about the shape and size of the pores, enabling a characterization of the permeability of the layer. The generated SymProFold models can also be used in combination with computational methods to analyze the impact of environmental factors such as pH on the assemblies, as well as to study interaction events of ligands or receptors to the S-layer.

#3.13

13. It would be interesting to know why the test set (19 annotated S-layer proteins (SLPs), three non-SLPs, and a viral capsid from the Odonata-associated circular virus 21) was chosen and how it was selected.

We appreciate the remark of reviewer #3. SymProFold originated from our aim to provide a detailed structural elucidation of the *Lactobacillus acidophilus* S-layer assembly and evolved into a general program for S-layer predictions, which we later extended to viral capsids. We did not have any predefined selection criteria.

First, we calculated S-layer assemblies for which experimental data was available (see Table 1) and included S-layers from all domains of life (Gram-positive and Gram-negative bacteria, as well as archaea). Next, we included calculations of annotated SLPs lacking experimental data. In specific cases, we observed that some annotated SLP genes were incorrect, resulting in the termination of the pipeline, and therefore had to be excluded from the analysis. After calculating numerous assemblies, the SymProFold pipeline was refined and improved, ultimately allowing for the prediction of viral capsids as well.

Although the pipeline is designed specifically for S-layer assemblies, it can also be extended to virus capsids. The Odonata-associated circular virus 21 capsid was selected as a benchmark because its capsid protein gene was annotated correctly, and the calculated assembly represents a good example of the program's utility.

Regarding the selection of the three non-SLPs, we wanted to present examples of 3-fold, 4-fold, and 6-fold rotational symmetry but without the repeating pattern (like present in the S-layer and virus capsids). For this purpose, we used proteins with known reported rotational symmetry in the literature.

#3.14

14. Page 16;: *‘whether the associated rotational symmetry axis is uniform within a tolerance range’. The specific tolerance range used in this step is not mentioned.*

The paragraph of the Methods section ‘Prediction of Oligomer Sets and Filtering’ was rephrased and a maximum deviation of 5Å was included:

“Therefore, the algorithm checks whether the occurring angles between the monomers correspond to a 2-, 3-, 4- or 6-fold rotational symmetry and whether the associated rotational symmetry axis is uniform within a tolerance range (max. of 5 Å deviation for each monomer).”

#3.15

15. *If the prediction cannot be performed using standard parameters of SymProFold, the authors refer to a ‘manual parameter setting’ of the pipeline (e.g. page 19; line 12). However, the process of performing this manual parameter setting is not clearly described.*

The referred sentence in the Method section "Prediction of the Viral Capsid" has been rewritten according to the annotation of reviewer #3 (see below). For the calculation of viral capsids, the SymProFold pipeline can be used to the step "Scoring of Symmetry Axis Complexes" (see Figure 1). The subsequent assembly of viral capsids is currently not provided by SymProFold and must be done manually.

“SymProFold currently does not support the automated assembly of viral capsids, but the results from the best scoring rotational symmetry axes can be manually built into an orientation that results in a fully assembled capsid.”

#3.16

16. *There appears to be a discrepancy in the cutoff values used for filtering predictions. For instance, one section mentions -"Only homodimer predictions with a minimal score of 0.3 $ipTM+pTM$ are further processed" , while another section states that "'Models with an*

ipTM+pTM score of ≥ 0.20 and existing interfaces between the monomers are treated as symmetry complexes.”

For the initial filtering steps, a more restrictive ipTM+pTM threshold is defined compared to subsequent processes. Specifically, during the prefiltering step for homodimer predictions, a ipTM+pTM threshold of 0.3 is set to ensure that only promising sequences enter the compute-intensive SymProFold pipeline. For the subsequent filtering step of the interfaces of the symmetry complexes, a less constraining ipTM+pTM threshold of 0.2 is used.

#3.17

17. The authors mention- “With respect to the complete protein chain, relaxed models with at least 3.0 clashes per 100aa (and 60.0 per 100aa for unrelaxed) are excluded. Additionally, in all possible sequence sections of length 200aa, only 6.0 clashes per 100aa for relaxed models (and 120.0 per 100aa for unrelaxed) are allowed”. Typically, the final structure should not exhibit any clashes involving main chain atoms.

We agree with reviewer#3. During the "Scoring of Symmetry Axis Complexes" step (see Figure 1), symmetry axis complexes (oligomer predictions) that exhibit extraordinarily high main chain and side chain clashes are filtered out. Currently, we do not use an additional filtering rule to explicitly exclude main chain clashes. Models predicted with a high ipTM+pTM score, resulting in good models, usually do not exhibit any main chain clashes. Typically, most clashes in the final assembly model do not originate from the symmetry axis complexes (“Prediction of multimer sets”), but from the superposition step. In this step (“Superposition and optimization”) the quality score ($\text{score}_{\text{quality}} = \text{score}_{\text{clash}} + \text{score}_{\text{bend}}$) is calculated. Clashes in the final model decrease the quality score and a careful revision of the final obtained model is advised.

#3.18

18. The ‘Computing requirements’ section lacks details about the settings for the prediction of the complexes, including parameters like the number of recycles.

We updated the paragraph "Computing Requirements" in the Methods section according to the remarks of reviewer #3 (see below).

“Computing requirements

An AlphaFold-Multimer 2.3 installation in standard configuration with full databases and v3 model weights was used for all complex calculations. The default value of max. 20 recycling iterations was used. The predictions were calculated at the VSC-5 Vienna Scientific Cluster (Vienna, Austria) on a GPU (NVIDIA A100, 40GB of vram).”

#3.19

19. The authors should add documentation to the cloud repository. Without this documentation, it was not feasible to test SymProFold.

We agree with reviewer#3 and added a detailed installation guide and guided tutorial on the GitHub repositories (<https://github.com/symprofold>). In the tutorial, the specific steps of the SymProFold pipeline are explained by using the example of the S-layer of *Vibrio aerogenes*. The code for the domain separator tool is also available separately on GitHub (https://github.com/symprofold/Domain_Separator) and includes a step-by-step instruction for the users.

Also see answer **#1.1** from Reviwer#1

REVIEWERS' COMMENTS

Reviewer #1 (Remarks to the Author):

The authors addressed most of the reviewers' comments. Moreover, the newly determined experimental structures strengthen the manuscript.

Regarding the MSA (comment #1.4), I recommend checking whether the success is correlated with the MSA depth, which considers the number and the diversity of sequences in the MSA. It is often measured by Neff (equation 1
<https://academic.oup.com/bioinformatics/article/36/4/1091/5556814#E1>)

Reviewer #1 (Remarks on code availability):

I browsed the repository but didn't try to install the code

Reviewer #2 (Remarks to the Author):

I would like to thank the authors who have fully addressed my initial comments.

Reviewer #2 (Remarks on code availability):

The code repository provides the README and the tutorial.

Review answers to “SymProFold: Structural prediction of symmetrical biological assemblies”
– resubmission 01.07.2024:

Reviewer #1

“Regarding the MSA (comment #1.4), I recommend checking whether the success is correlated with the MSA depth, which considers the number and the diversity of sequences in the MSA. It is often measured by Neff (equation 1 <https://academic.oup.com/bioinformatics/article/36/4/1091/5556814#E1>)”

We thank Reviewer #1 for this remark. In response, we calculated the Neff values for all subchains used in this study, which contributed to the successful assemblies. We have included a new Supplementary Table 7 that provides the integrated Neff values for all subchains. Additionally, we added a brief paragraph to the "Assessment of Prediction Quality" section (see below). Our analysis did not reveal any correlation between high (or low) Neff values and successful outcomes.

“The number of effective sequences (Neff) in the MSA for the individual subchains was calculated using NEFFy, which did not correlate with a successful outcome. Subchains with an integrated Neff of below 5 still can result in an assembled layer (Supplementary Table 7).”